# Structural basis for the activation of the lipid scramblase TMEM16F

**Melanie Arndt** [1,3], **Carolina Alvadia** [1,3], **Monique S. Straub** [1], **Vanessa Clerico Mosina** [2], **Cristina Paulino** [2] **& Raimund Dutzler** [1] ✉

TMEM16F, a member of the conserved TMEM16 family, plays a central role in the initiation of blood coagulation and the fusion of trophoblasts. The protein mediates passive ion and lipid transport in response to an increase in intracellular $Ca^{2+}$. However, the mechanism of how the protein facilitates both processes has remained elusive. Here we investigate the basis for TMEM16F activation. In a screen of residues lining the proposed site of conduction, we identify mutants with strongly activating phenotype. Structures of these mutants determined herein by cryo-electron microscopy show major rearrangements leading to the exposure of hydrophilic patches to the membrane, whose distortion facilitates lipid diffusion. The concomitant opening of a pore promotes ion conduction in the same protein conformation. Our work has revealed a mechanism that is distinct for this branch of the family and that will aid the development of a specific pharmacology for a promising drug target.

The lipid scramblase TMEM16F facilitates the passive diffusion of phospholipids between both leaflets of the membrane[1]. The protein is inactive under resting conditions and becomes activated in response to an increase of the intracellular $Ca^{2+}$ concentration. Scrambling causes the dissipation of the lipid asymmetry of the plasma membrane leading to the exposure of the negatively charged phosphatidylserine (PS) to the outside, which in a resting state is confined to the inner membrane leaflet[2,3]. The exposed PS is recognized by receptors in a context-dependent manner to trigger different physiological processes. In platelets, PS exposure leads to the assembly of the prothrombin complex, which initiates the blood coagulation cascade[3,4]. Mutations of TMEM16F interfering with its expression have been associated with bleeding disorders[5]. TMEM16F activity was also linked to pyroptosis[6] and bone mineralization[7]. In trophoblasts, TMEM16F-mediated lipid scrambling promotes syncytialization[8] whereas its close homologue TMEM16E facilitates fusion of muscle precursor cells[9]. Recently, TMEM16F-mediated cell fusion was identified as a critical step in viral infections and its inhibition was proposed as a strategy to prevent virus-induced syncytia formation[10]. TMEM16F is a member of the conserved TMEM16 family, which is restricted to eukaryotes and encompasses ten paralogs in humans[11–13]. Whereas TMEM16A and TMEM16B function as calcium-activated chloride channels that do not show any lipid transport activity[14–16], the remaining eight paralogs (i.e. TMEM16C-K) are supposed to function as lipid scramblases[11,17]. Scrambling is best understood for fungal homologs based on structures determined by X-ray crystallography[18] and cryo-electron microscopy (cryo-EM)[19–22]. Fungal TMEM16 structures have defined the general architecture of the family, where subunits, consisting of an N-terminal cytoplasmic domain followed by a membrane-inserted part composed of ten transmembrane helices (α1-10), assemble as homodimers[18]. The subunits are functionally independent[23,24], each containing an extended site that facilitates lipid diffusion termed 'subunit cavity'. Constituted by α-helices 3–7, the subunit cavity is located at the periphery of the protein, opposite to the dimer interface[18]. Its access to the membrane is regulated by $Ca^{2+}$ binding to a proximal site that is embedded within the inner membrane leaflet. There, the binding of two $Ca^{2+}$ ions causes the rearrangement of the peripheral helix (α6), which contributes residues for their coordination and also lines the presumed lipid permeation path[20,25]. This rearrangement triggers a larger conformational change in the subunit cavity, leading to the dissociation of mutual interactions between α-helices 4 and 6 and the opening of a furrow that offers a continuous

[1]Department of Biochemistry University of Zurich, Winterthurer Str. 190, CH-8057 Zurich, Switzerland. [2]Department of Structural Biology and Membrane Enzymology at the Groningen Biomolecular Sciences and Biotechnology Institute, University of Groningen, Nijenborgh 4, 9747 AG, Groningen, The Netherlands. [3]These authors contributed equally: Melanie Arndt, Carolina Alvadia. ✉e-mail: dutzler@bioc.uzh.ch

hydrophilic pathway for the diffusion of lipid headgroups across the hydrophobic core of the bilayer[18]. This process, where the hydrophobic fatty acid chains remain embedded in the membrane, resembles the 'credit card mechanism' of lipid transport that was previously proposed based on theoretical considerations[26]. In addition, TMEM16 scramblases distort the surrounding membrane environment thereby lowering the barrier for lipid movement and channeling lipids into the subunit cavity[20,25,27,28]. A similar mechanism was also found for the human phospholipid scramblase TMEM16K[29]. Conversely, the equivalent activation step in the channel TMEM16A does not lead to the opening of a membrane-accessible cavity but instead promotes the activation of a selective anion conduction pore[30,31].

With respect to transport, TMEM16F combines properties of both functional branches of the family. Whereas its known physiological role is associated to lipid scrambling[1,32,33], it was also characterized as an ion channel of low selectivity that poorly discriminates between anions and cations and that was also proposed to be permeable for $Ca^{2+}$ (ref. [33], [34]). Ion and lipid conduction are activated by the same $Ca^{2+}$ binding step and thus appear to be mechanistically related[19]. In their tight regulation, both functional properties of TMEM16F appear to be distinct from equivalent processes in fungal scramblases, which show considerable scrambling activity in absence of $Ca^{2+}$ (ref. [18], [35]) and where scrambling-associated ion permeation was solely detected upon their reconstitution in proteoliposomes[35,36]. Known structures of TMEM16F[19,37] strongly resemble the channel TMEM16A[30,38], with which the protein shares a close evolutionary relationship. Even in presence of $Ca^{2+}$, the cavity does not show pronounced conformational rearrangements leading to its opening[19,37]. It was thus questioned whether such opening would at all be necessary to activate lipid scrambling in TMEM16F[37], particularly since it was proposed for fungal scramblases that scrambling could proceed distant from the hydrophilic furrow due to protein-induced membrane destabilization[22,39]. However, a different study has identified residues buried in the closed subunit cavity of TMEM16F, whose mutation have caused a strong enhancement of scrambling activity[40], suggesting that activation of TMEM16F also relies on a larger conformational rearrangement of this region of currently unknown nature.

Although previous studies have defined the molecular architecture of TMEM16F and revealed its dual function as ion channel and lipid scramblase[19,32,34,37], they provided little mechanistic insight into the structural features that facilitate ion and lipid transport. To clarify these open questions, we here characterize the functional properties of mutants of residues on α-helices 4–6, which together constitute the presumed ion and lipid conduction pathway. Our study provides a comprehensive overview of the role of pore-lining amino acids for the stabilization of distinct states of TMEM16F and allows the identification of positions where mutations stabilize the active state of the protein. These positions largely conform with a previous study postulating an inner gate for lipid scrambling[40]. The subsequent structure determination of activating mutants in detergent and lipid nanodiscs reveals pronounced $Ca^{2+}$-induced conformational changes that were not observed in the WT protein, and which lead to a reorganization of the subunit cavity to facilitate ion and lipid diffusion.

## Results

### Scanning mutagenesis of the pore region

At the onset of our study, we were interested in the relevance of residues located on α-helices 4–6, which are buried in the common core of the presumably closed permeation path in WT structures, for conformational transitions that facilitate lipid scrambling and ion conduction. To this end, we characterized alanine mutants with regard to both functional properties (Fig. 1, Supplementary Fig. 1). We investigated the effect of mutations on lipid conduction using a cellular scrambling assay. Protein constructs were expressed in a cell line carrying a genetic knock-out of TMEM16F (generously provided by Dr.

Huanghe Yang), which itself lacks any $Ca^{2+}$-dependent scrambling activity[41]. In these experiments, we quantified the steady-state fluorescence resulting from the binding of labeled annexin V to exposed phosphatidylserine at resting $Ca^{2+}$-concentrations and the fluorescence increase 780 s after the addition of ionomycin, which elevates the intracellular $Ca^{2+}$ level (Fig. 1a, Supplementary Fig. 1a). This assay, which is particularly suited for the detection of mutants with strongly activating phenotype that show elevated fluorescence compared to WT already at resting $Ca^{2+}$ concentrations, revealed increased activities for Phe 518 on α4, Asn 562 and Tyr 563 on α5, and Gln 608 and Ile 612 on α6 (Fig. 1a, b, Supplementary Fig. 1a). In complementary studies based on electrophysiology, we expected mutations that stabilize an activated state to increase, and conversely, ones that stabilize the resting state to lower $Ca^{2+}$-potency. Such behavior was observed in a previous investigation of the chloride channel TMEM16A[31]. Our studies revealed a predominant left-shift of the $EC_{50}$ for positions on α4 and a right-shift for several mutants on α6, that are located extracellular to the $Ca^{2+}$-binding site (Fig. 1a–d, Supplementary Fig. 1b). Notable exceptions on α6 concern Ile 611, Ile 612 and Gln 623, which are positioned on both sides in relation to Gly 615, the pivot for the rearrangement of α6 in response to $Ca^{2+}$-binding[19] (Fig. 1a, Supplementary Fig. 1b). The magnitude of the shifts varies in size, deviating up to 35% from the original values when plotted on a logarithmic scale (Fig. 1a). Differences in both directions are found for mutations on α5, which constitutes the center of the subunit cavity[18]. On this helix, point mutations of residues Asn 562 and Tyr 563 result in a two-fold decrease of the $EC_{50}$, revealing two positions where sidechain truncations strongly destabilize the closed state (Fig. 1a, b, d).

In combination, both assays revealed a similar picture of the energetic contributions of residues to the activation of TMEM16F. Most mutations affected both of its functions as ion channel and lipid scramblase in a similar manner, with mutation of a group of residues located close to the center of the membrane to alanine exerting a strong stabilization of the open state (Fig. 1b–d). Several of these residues conform with positions identified in a recent study based on molecular dynamics simulations and cellular lipid transport assays that were assigned a function as intracellular gate[40].

### Functional properties of activating mutants

After identifying positions where sidechain truncations enhance activation, we became interested in how the mutants act to stabilize the open state of TMEM16F. To this end, we have selected the residues Phe 518 on α4 and Asn 562 and Tyr 563 on α5 for more detailed investigations. For all three positions, we have systematically studied point mutants with distinct sidechain properties to correlate the effect of the mutation on protein function. Initially, we characterized ion conduction properties by patch-clamp electrophysiology after replacing each of the three positions with amino acids of varying hydrophilic character (Fig. 2a–d). For the α4 residue Phe 518, we have investigated six mutants ranging from the aliphatic isoleucine to the polar histidine (Fig. 2a, b). Among mutants of this position, we find a linear correlation of the $EC_{50}$ with the hydrophilicity of the sidechain replacement with histidine exerting the strongest effect, resulting in a three-fold decrease of the $EC_{50}$ (i.e., from 3 to 1 μM, Fig. 2b). Remarkably, this correlation was only observed for Phe 518 and not for the two α5 residues Asn 562 and Tyr 563, where the truncation to alanine had the largest impact on calcium potency (Fig. 2c, d). A similar phenotype of Phe 518 mutants was found in the cellular scrambling assay, where the replacement with polar sidechains increased scrambling activity at both, resting and elevated $Ca^{2+}$ concentrations (Fig. 2e, f). Finally, we investigated the $Ca^{2+}$ dependence of lipid scrambling of reconstituted mutants of Phe 518 in vitro with an assay that was previously used to determine $Ca^{2+}$-concentration relationships for lipid conduction of WT TMEM16F[19] (Supplementary Fig. 2). In these experiments, we found a concordant

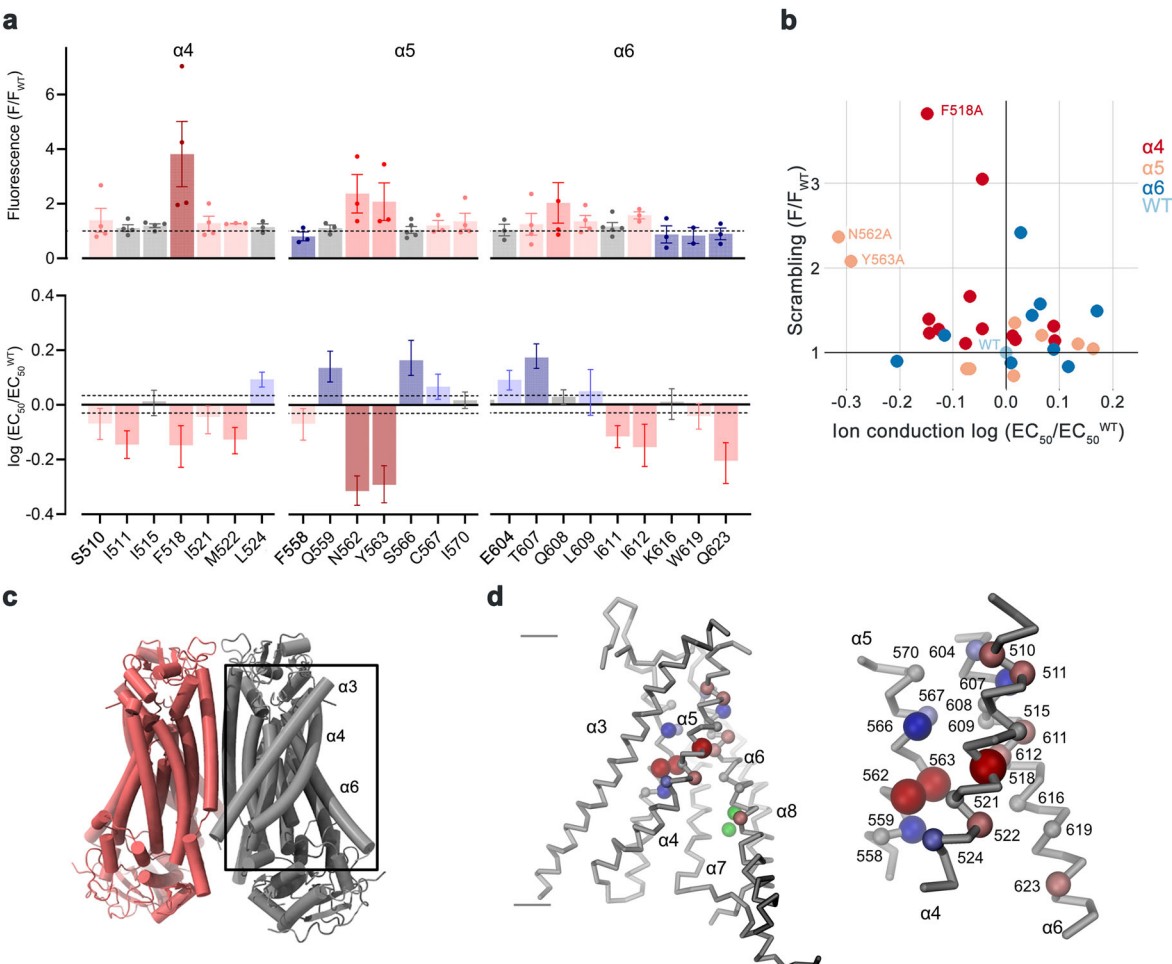

**Fig. 1 | Activation properties of pore mutants. a** Alanine scan of residues lining the pore of the closed ion and lipid permeation region of TMEM16F. Top, initial fluorescence in a cell-based lipid scrambling assay monitoring the binding of fluorescently tagged annexin V to phosphatidylserine at the cell surface. Values reflecting the lipid transport activities of TMEM16F mutants at resting $Ca^{2+}$-concentrations were normalized to the fluorescence of WT (dashed line). Data show mean of the displayed number of experiments, errors are s.e.m. Bottom, $EC_{50}$ of $Ca^{2+}$ activation recorded in excised patches. Data show $EC_{50}$ values derived from a Hill-fit of dose-response curves shown in Supplementary Fig. 1b. Mutant $EC_{50}$s are expressed as log-ratio compared to WT, errors are 95% confidence intervals. Dashed lines refer to the 95% confidence interval of WT. Colors reflect direction and magnitude of change. **b** Scatter plot illustrating the relationship between mean values for scrambling normalized to WT and log-fold changes in the $EC_{50}$ for each mutant as depicted in **a**. Colors refer to the location of the site of mutation. Data from selected residues are labeled. **c** General architecture of TMEM16F (PDBID 6QPB). The box highlights the subunit cavity. **d** Cα representation of the subunit cavity constituting the ion and lipid permeation region of TMEM16F (PDBID 6QP6) with Cα positions of mutated residues shown as spheres and colored according to the magnitude of the effect shown in **a**. Membrane boundaries are indicated. Inset (right) shows blowup of the tightly packed region with residue number of mutated sites indicated. Source data are provided as a Source Data file.

$Ca^{2+}$-dependence of scrambling as observed for ion conduction with the mutant F518H showing the strongest activating properties among the investigated constructs (Fig. 2g, h, Supplementary Fig. 2). Collectively, our experiments have demonstrated that the replacement of Phe 518 by hydrophilic residues exerts a strongly stabilizing effect on the active state of TMEM16F, which extends to both ion- and lipid conduction. As the residue is located at the interface between α-helices 4 and 6, these results also emphasize the role of a functional crosstalk between both helices during $Ca^{2+}$-activation. F518, Y563 and N562 cluster within the narrowest part of the presumed region of ion and lipid permeation. Consequently, the functional phenotype of their mutation hints towards a conformational change in this region that takes place during activation and that was not observed in the $Ca^{2+}$-bound structures of WT[19,37].

### Structures of activating mutants
In the next step, we took advantage of the identification of activating mutants to capture conformational changes in TMEM16F that escaped

detection in previous structures, due to the high stability of closed states under the investigated conditions. We thus set out to study the mutant F518H by cryo-electron microscopy (cryo-EM) and determined structures of the $Ca^{2+}$-free protein in detergent and the $Ca^{2+}$-bound protein in both detergent and lipid nanodiscs (Fig. 3, Supplementary Figs. 3–5, Table 1). All structures are of high quality and have provided detailed insight into accessible conformations of TMEM16F. The structure of F518H in detergent in absence of $Ca^{2+}$ (F518H$^{noCa}$) determined at 3.39 Å defines a protein conformation that is very similar to the corresponding structure of WT (WT$^{noCa}$, PDBID 6QPB, RMSD 0.64 Å)[19] (Fig. 3a, b, Supplementary Figs. 3 and 6a). Compared to the previously determined $Ca^{2+}$-bound WT structure (WT$^{Ca}$, PDBID 6QP6)[19], the intracellular part of α6 has detached from the vacant $Ca^{2+}$-binding site by a rigid-body rotation around Gly 615 serving as a pivot for the displacement (resulting in an RMSD of 2.04 Å) (Fig. 3a–c, Supplementary Fig. 6a, b). In the three structures depicting presumable inactive states (i. e., WT$^{noCa}$, WT$^{Ca}$, and F516H$^{noCa}$), α3 and α4 are tightly packed against the core of the protein and engaged in

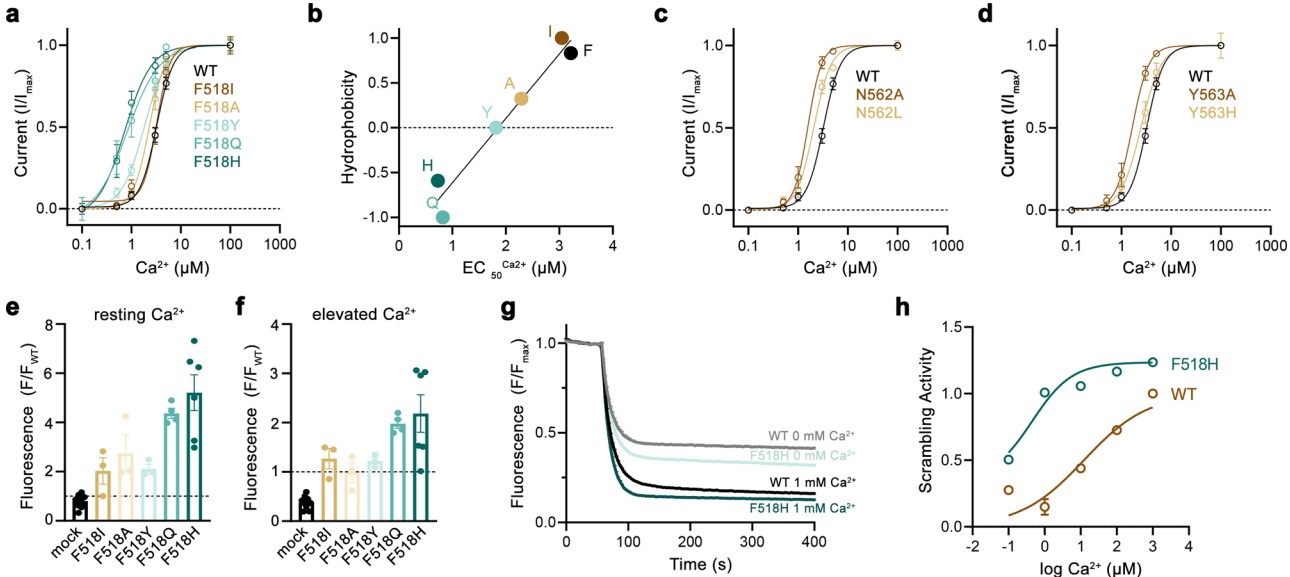

**Fig. 2 | Characterization of activating mutants. a** $Ca^{2+}$-concentration-response relationships of mutants of Phe 518 measured in inside-out patches. **b** $EC_{50}$ values of $Ca^{2+}$ plotted against the normalized hydrophobicity (Eisenberg and Weiss Scale) of the respective Phe 518 mutations. Line shows a fit to the data. **c** $Ca^{2+}$-concentration-response relationships of mutants of Asn 562 and **d** Tyr 563 measured in inside-out patches. **a, c, d** Data show averages of multiple experiments derived from independent cells (WT $n = 10$, F518H $n = 5$, F518Q $n = 4$, F518Y $n = 5$-6, F518A $n = 5$-9, F518I $n = 7$, N562A, $n = 5$, N562L $n = 4$, Y563A $n = 5$, Y563H $n = 3$–6), errors are s.e.m., lines show fit to a Hill equation. **e, f** Lipid transport of Phe 518 mutants quantified in a cellular scrambling assay. **e** Initial values recorded at resting $Ca^{2+}$ concentration and **f** levels measured 600 s after application of ionomycin, which increases intracellular $Ca^{2+}$. Individual experiments are depicted as spheres (mock $n = 10$, F518I, F518A, F518Y $n = 3$, F518Q $n = 4$, F518H $n = 6$), errors are s.e.m. **g, h** Liposome-based in vitro scrambling assay of the reconstituted mutant F518H in comparison to WT. **g** Time-dependent fluorescence decrease upon addition of the reducing agent dithionite (t = 60 s) at 0 and 1000 μM $Ca^{2+}$ compared to WT. Data show mean of three technical replicates, errors (s.e.m.) are smaller than the displayed line width. **h** $Ca^{2+}$-concentration response relationship of scrambling of the mutant F518H compared to WT obtained from three technical replicates (displayed in Supp. Fig. 2d). Values were obtained as described in the methods. Solid line shows fit to a Hill equation, errors are s.e.m. Source data are provided as a Source Data file.

numerous interactions thereby sealing the assumed ion and lipid permeation path from the environment (Fig. 3a–c).

In contrast to F518H$^{noCa}$, the two structures of the $Ca^{2+}$-bound mutant, determined at 2.96 Å in detergent in presence of the lipid $PIP_2$ (F518H$^{Ca}$, Supplementary Fig. 4, Table 1) and at 2.93 Å in lipid nanodiscs (F518H$^{Ca}_{ND}$, Supplementary Fig. 5, Table 1), adopt conformations that are distinct from WT$^{Ca}$ (RMSDs 2.0 Å for F518H$^{Ca}_{ND}$ and 2.88 Å for F518H$^{Ca}$) with both structures carrying characteristics of an activated state (Fig. 3d–i, Supplementary Fig. 6b). The most pronounced differences compared to WT$^{Ca}$ concern α-helices 3 and 4 and to a lesser extent α6 and its preceding loop region (Fig. 3f, i, Supplementary Fig. 6b). In this transition, the F518H$^{Ca}_{ND}$ structure shows an apparent intermediate towards the structure of the mutant in detergent since its coordinates are about halfway on a potential trajectory from F518H$^{noCa}$ to F518H$^{Ca}$ (Fig. 4a–c). In F518H$^{Ca}_{ND}$, the poor resolution of the α3-α4 pair compared to the rest of the protein reflects its high mobility under the investigated conditions, whereas in F518H$^{Ca}$ both helices are found in a single well-defined conformation (Fig. 3d, g, Supplementary Figs. 5, 6). In both ligand-occupied mutant structures, the extracellular part of the α3-α4 pair has detached from the rest of the protein, thereby unleashing its tight interactions with α5 and α6 (Fig. 3f, i and 4d). In case of α4, this leads to the straightening of the bent conformation adopted in WT$^{Ca}$ by 16° in F518H$^{Ca}_{ND}$ and 26° in F518H$^{Ca}$ (Fig. 4b). In both structures, the relaxation of the bent conformation of α4 is accompanied by a tilt of the helix and a shift by 3 Å along its axis towards the cytoplasm (Fig. 4a, b). The concomitant conformational change of α3 from WT$^{Ca}$ to F518H$^{Ca}_{ND}$ can be approximated by comparable rearrangements. The tilt around an axis located in the center of the helix parallel to the membrane leads to an outward movement of α3 by 25° at its extracellular and 10° at its intracellular part and a shift along its axis towards the cytoplasm (Fig. 4c). Compared to F518H$^{Ca}_{ND}$, the intracellular part of α3 in F518H$^{Ca}$ is preserved, but there is an additional large

conformational change in proximity of Gly 473, where the helix has unwound and its extracellular part has rotated as rigid unit by 46° towards the membrane core to assume a conformation that is stabilized by contacts with the straightened α4 (Figs. 3i and 4a). As a consequence of the described transitions, the interaction interface between α4 and α6 decreases from 747 Å$^2$ in WT$^{Ca}$ to 535 Å$^2$ in F518H$^{Ca}_{ND}$ and 241 Å$^2$ in F518H$^{Ca}$. In the latter structure, the remaining contacts involve interactions of the mutated His 518 on α4 with Trp 619 and Gln 623 on α6, which are distant from each other in the WT structure (Fig. 4e, f). The described conformational changes result in the opening of a hydrophilic pore at the interface of α helices 3–6 with a diameter of 5 Å at its constriction, which would be of sufficient size to permit conduction of ions that have stripped a large part of their hydration shell (Fig. 5a–c, Supplementary Fig. 6c, d). The same transition exposes polar residues at the upper part of the subunit cavity to the membrane, which were covered in the core of the protein in the apo structure of the mutant and apo and $Ca^{2+}$-bound structures of WT (Fig. 5d, e, Supplementary Fig. 6d). Although we cannot exclude the possibility that in a fully scrambling-competent conformation, residual contacts between α4 and α6 would be broken to open a subunit cavity that exposes its inside to lipids for the entire thickness of the membrane, as observed in fungal TMEM16 scramblases, we did not find any evidence for such conformation in our data. We thus suppose that the observed $Ca^{2+}$-occupied F518H structures are capable of catalyzing lipid permeation. In this case, lipids would during their transition across the membrane surmount the contact region between α-helices 4 and 6 outside of the subunit cavity, as described as potential scenario in previous studies[19,22].

To investigate whether equivalent conformational changes, as found for the mutant F518H, would also be observed in activating mutants that are located remote from the α4-α6 interface, we have determined the structure of the mutant N562A. This residue is located on α5 and does not line the predicted pore. A dataset of the

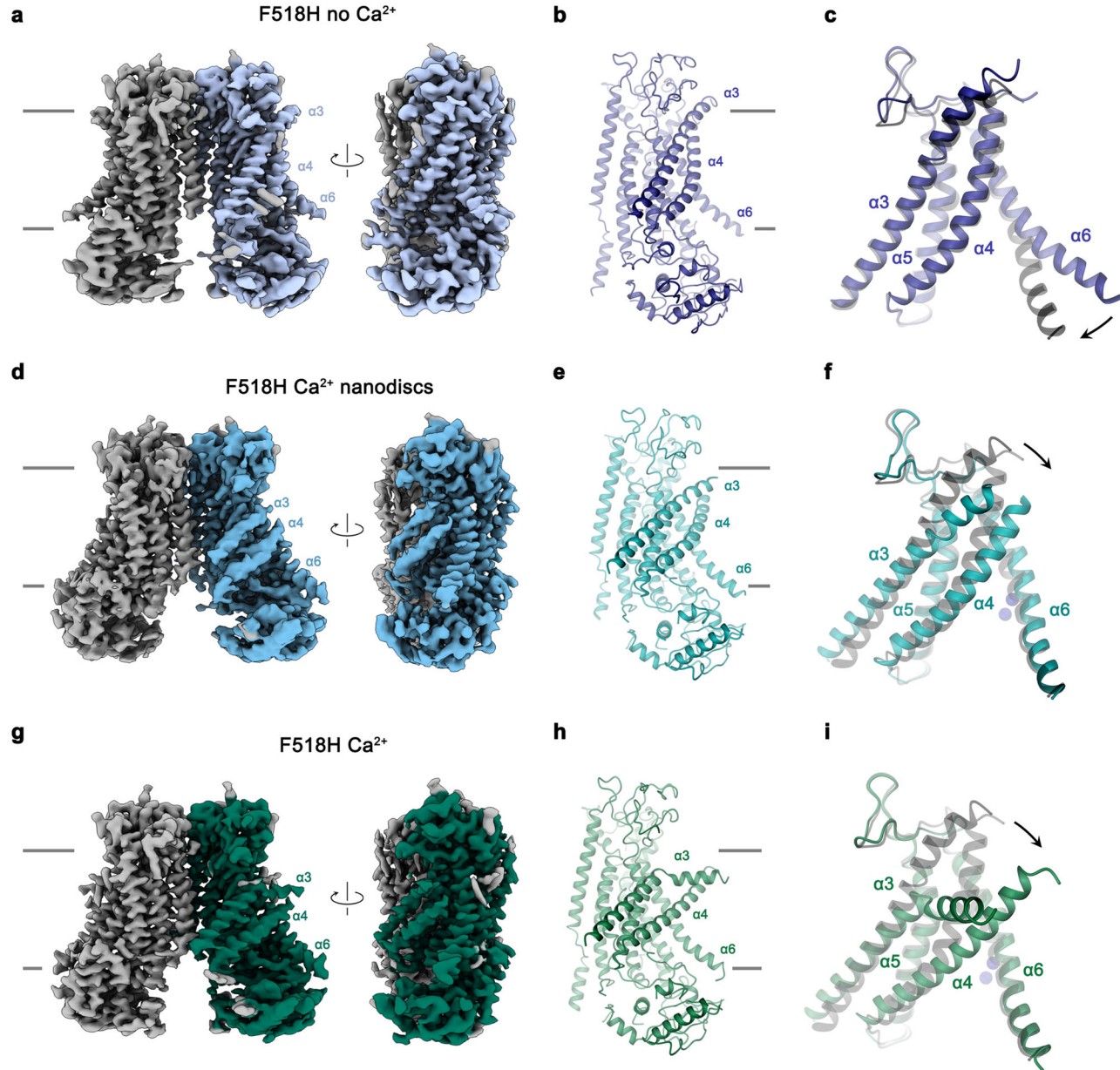

**Fig. 3 | Structures of the mutant F518H. a** Cryo-EM density of the F518H$^{noCa}$ structure at 3.39 Å with one subunit shown in color. **b** Ribbon representation of the F518H$^{noCa}$ subunit and **c** blowup of the pore region. **d** Cryo-EM density of the F518H$^{Ca}_{ND}$ structure at 2.93 Å with one subunit shown in color. **e** Ribbon representation of the F518H$^{Ca}_{ND}$ subunit and **f** blowup of the pore region. **g** Cryo-EM density of the F518H$^{Ca}$ structure at 2.96 Å with one subunit shown in color. **h** Ribbon representation of the F518H$^{Ca}$ subunit and **i** blowup of the pore region. **a, d, g** Relationships between views are indicated. **a, b, d, e, g, h** Lines indicate membrane boundaries. **c, f, i** WT$^{Ca}$ (PDBID 6QP6) is shown in gray for comparison, arrows indicate movements. **a–i** Selected transmembrane helices are labeled. **f, i** Ca$^{2+}$ ions are displayed as blue spheres.

mutant in detergent, in presence of 2 mM Ca$^{2+}$ and 0.01 mM of the lipid PIP$_2$ (N562A$^{Ca}$), allowed a reconstruction of high quality, showing two populations of the protein obtained after a 3D variability analysis and refinement of the two end states (Supplementary Fig. 7, Table 1). One population, containing 62% of the classified particles, displays C2 symmetry and was refined to 3.01 Å (Supplementary Fig. 7c–f). It shows the protein in a non-conducting closed conformation (c) that is essentially identical to the structure of WT$^{Ca}$ determined under equivalent conditions (Fig. 6a). A second population encompassing 38% of the classified particles, refined to 3.49 Å, is asymmetric with one subunit residing in the same non-conducting conformation observed in the symmetric class and the other subunit showing an arrangement that closely resembles the structure of the

mutant observed in the dataset F518H$^{Ca}$ (o), displaying similar movements of α-helices 3, 4 and 6 (Fig. 6b–d, Supplementary Fig. 7c, g–j). As in F518H$^{Ca}$, α3 and α4 have rearranged towards the opening of the subunit cavity although the extracellular parts of both α-helices are mobile and thus not defined in the density (Fig. 6b–d, Supplementary Fig. 7i, j). Consequently, there is no direct evidence for a kinked conformation of α3 in this dataset. Assuming a binomial distribution of states in subunits that act independently in the dimeric protein, as described for the anion channel TMEM16A[23,24], the observed ratio suggests that the large predominance of the closed state in WT was perturbed in the mutant, where 20% of subunits reside in an activated conformation. The structure of the N562A mutant thus confirms that the rearrangement of the subunit cavity in

**Table 1 | Cryo-EM data collection, processing, refinement and validation statistics**

| | F518H$^{Ca}$ (EMD-15914) (PDB 8B8J) | F518H$^{noCa}$ (EMD-15913) (PDB 8B8G) | F518H$^{Ca}_{ND}$ (EMD-15919) (PDB 8B8Q) | N562A$^{Ca}_{cc}$ (EMD-15916) (PDB 8B8K) | N562A$^{Ca}_{oc}$ (EMD-15917) (PDB 8B8M) |
|---|---|---|---|---|---|
| *Data collection and processing* | | | | | |
| Magnification | 130.000 | 130.000 | 130.000 | 130.000 | 130.000 |
| Voltage (kV) | 300 | 300 | 300 | 300 | 300 |
| Electron exposure (e–/Å$^2$) | 66.6/68.1 | 69.8 | 76 (UltrAuFoil tilt)/ 84.2 (UltrAufoil)/66.6 (GO) | 62.4 | 62.4 |
| Defocus range (μm) | −1 to −2.4 | −1 to −2.4 | −1 to −2.4 | −1 to −2.4 | −1 to −2.4 |
| Pixel size (Å) | 1.302 | 1.302 | 1.302 | 1.302 | 1.302 |
| Symmetry imposed | C2 | C2 | C1 | C2 | C1 |
| Initial particle images (no.) | 1,600,461 | 2,901,588 | 3,399,184 | 1,168,595 | 1,168,595 |
| Final particle images (no.) | 394,821 | 181,753 | 282,719 | 239,346 | 124,276 |
| Map resolution (Å) | 2.96 | 3.39 | 2.93 | 3.01 | 3.49 |
| FSC threshold | 0.143 | 0.143 | 0.143 | 0.143 | 0.143 |
| Map resolution range (Å) | 2.7–4.0 | 2.9–5.0 | 2.8–5.0 | 2.8–3.5 | 3.0–5.0 |
| *Refinement* | | | | | |
| Initial model used (PDB code) | 6QP6 | 6QPB | 6QP6 | 6QP6 | 6QP6 |
| Model resolution (Å) | 3.1 | 3.6 | 3.1 | 3.0 | 3.7 |
| FSC threshold | 0.5 | 0.5 | 0.5 | 0.5 | 0.5 |
| Model resolution range (Å) | 2.8–3.1 | 3.2–3.6 | 2.8–3.1 | 2.7– 3.0 | 3.2– 3.7 |
| Map sharpening *B* factor (Å$^2$) | 108.3 | 112.9 | 85 | 97.7 | 78.5 |
| Model vs. Map CC (mask) | 0.76 | 0.82 | 0.78 | 0.81 | 0.76 |
| Model composition | | | | | |
| Non-hydrogen atoms | 11,686 | 12,086 | 11,370 | 12,324 | 11,758 |
| Protein residues | 1418 | 1464 | 1382 | 1482 | 1423 |
| Ligands | 6 Ca$^{2+}$ | | 6 Ca$^{2+}$ | 6 Ca$^{2+}$ + 2 P1O | 6 Ca$^{2+}$ |
| *B* factors (Å$^2$) | | | | | |
| Protein | 136.63 | 114.71 | 120.20 | 113.15 | 109.84 |
| Ligand | 124.86 | | 109.53 | 112.40 | 99.04 |
| R.m.s. deviations | | | | | |
| Bond lengths (Å) | 0.003 | 0.004 | 0.003 | 0.003 | 0.003 |
| Bond angles (°) | 0.464 | 0.556 | 0.486 | 0.500 | 0.532 |
| Validation | | | | | |
| MolProbity score | 1.53 | 2.52 | 2.50 | 1.85 | 2.26 |
| Clashscore | 6.85 | 10.96 | 9.87 | 6.75 | 9.98 |
| Poor rotamers (%) | 1.25 | 7.59 | 7.36 | 2.92 | 4.46 |
| Ramachandran plot | | | | | |
| Favored (%) | 97.54 | 95.75 | 95.32 | 97.36 | 96.32 |
| Allowed (%) | 2.46 | 4.25 | 4.68 | 2.64 | 3.68 |
| Disallowed (%) | 0.00 | 0.00 | 0.00 | 0.00 | 0.00 |

F518H$^{Ca}$ and F518H$^{Ca}_{ND}$ is not a consequence of the disturbed interface of α-helices 4 and 6, but instead defines a structural transition towards the active state that is adopted upon Ca$^{2+}$-binding. The absence of asymmetric conformations in datasets of F518H obtained under equivalent conditions (Supplementary Figs. 4 and 5) further emphasizes the gradual stabilization of the active state, which is more pronounced in the case of the mutant of the residue on α4 than observed for N562A.

**Features of the detergent and lipid layers surrounding TMEM16F**
In all structures determined in this study, features of the density surrounding the protein, which depending on the preparation correspond to either detergents or lipid nanodiscs, further illustrate the effect of the observed structural rearrangement on its environment. Whereas this density is regular in the Ca$^{2+}$-free conformation of F518H

and around subunits in both populations of N562A where the subunit cavity has remained closed, there is a pronounced attenuation at the site of conformational changes in subunits showing an activated state in both detergent and lipid nanodiscs (Fig. 7). These anomalies in the density distribution point towards a destabilization of the lipid and detergent environment. As membrane thinning and bilayer distortion was previously considered as a hallmark for lipid scrambling[20,22,25,37], the association of these features with the activated state suggests that the observed rearrangements are relevant for the scrambling activity of TMEM16F.

**Functional relationships between active state mutants**
Despite the described differences regarding conformational details, the structures of the activating TMEM16F mutants F518H in detergent and lipid nanodiscs and N562A in detergent all show common

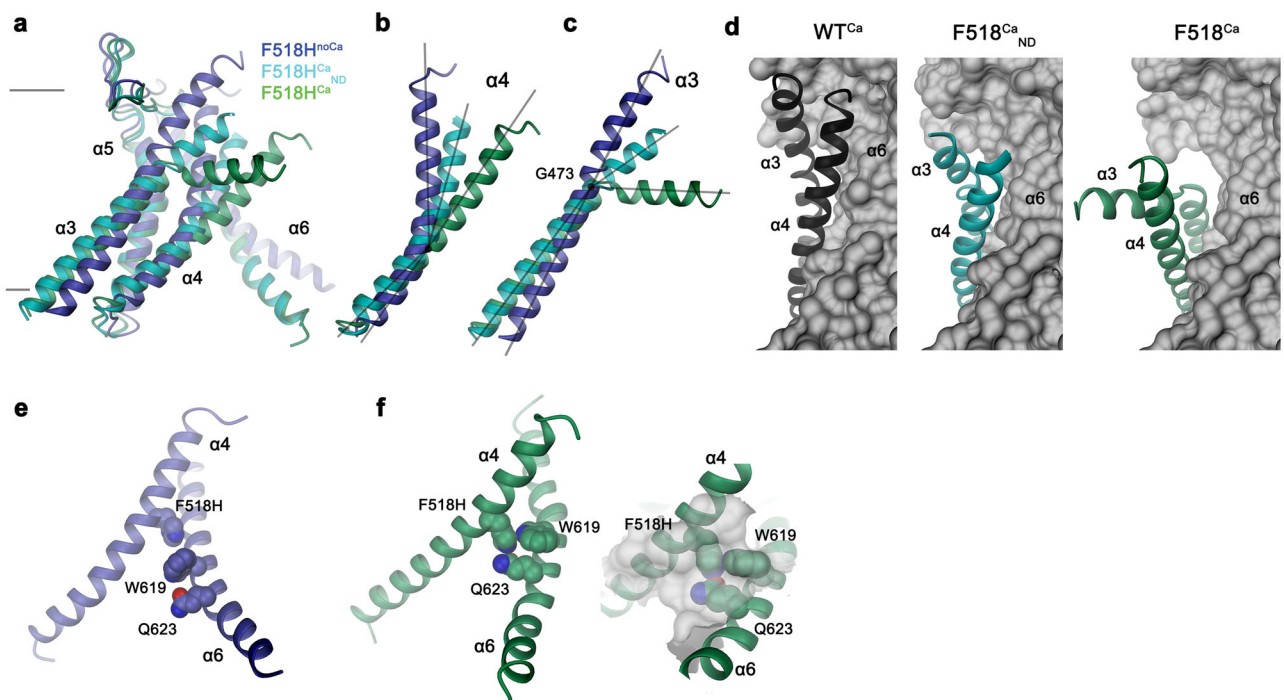

**Fig. 4 | Structural features of the activating mutant F518H.** Superposition of the pore region of different structures of F518H encompassing α-helices 3-6 (**a**), α4 (**b**) and α3 (**c**). **b, c** Lines indicate axes for different helix sections to approximate conformational changes. **d** Packing interactions of α3 and 4 (ribbon) with the remainder of the protein (depicted as molecular surface) in different conformations of TMEM16F. Relationship between α4 and α6 in **e**, F518H[noCa] and **f**, F518H[Ca] with inset displaying blow-up of the contact region with sections of the molecular surface shown.

rearrangements towards the opening of the subunit cavity in the Ca²⁺-bound state. These are characterized by the concerted movement of α-helices 3 and 4 away from the core of the protein that is accompanied by a straightening of the bent α4 and its dissociation from α6 (Fig. 3f, i and 4a–d). To further investigate the relevance of the observed conformational changes for activation, we have studied combinations of point mutants of involved residues by patch-clamp electrophysiology and cellular scrambling assays.

One of the striking features of F518H[Ca] concerns a conformational change in the extracellular part of α3, resulting from its local unwinding close to Gly 473, a residue that is conserved in TMEM16A and F (Fig. 3h, i, 4c and 8a). We constructed mutants of this flexible residue with the aim to rigidify this region and investigated the consequence of the mutation on activation. The mutation of the equivalent position (i.e., Gly 510) in TMEM16A to a rigid proline, investigated in a previous study, has shown a pronounced right-shift of the EC₅₀ of Ca²⁺, reflecting a strong destabilization of the open state[42]. The concomitant weakening of its inhibition by an open-channel blocker, which binds to a close-by pocket that is formed in the active state, further underlines the restriction of conformational rearrangements by the mutation[42]. The equivalent substitution of Gly 473 in TMEM16F to proline (G473P) exerts an even more pronounced effect and did not show any visible activity in cellular scrambling assays and patch-clamp recordings, despite its targeting to the plasma membrane (Fig. 8b–d, Supplementary Fig. 8a). By contrast, a mutation of the same residue to alanine (G473A) had no detectable impact on its functional behavior (Fig. 8b, Supplementary Fig 8a, b). These results illustrate the importance of the described position for structural rearrangements in TMEM16F underlying its activation.

The second pronounced feature observed in all activated conformations concerns the rearrangement of α4 leading to its dissociation from α6 and the formation of novel interactions between both helices. These involve contacts between the residue at position 518 with Trp 619 and Gln 623 (Fig. 4e, f). The latter residues are 6 and 11 Å apart from His 518 in F518H[noCa], precluding a direct interaction in the absence of Ca²⁺ (Fig. 4e, and 8e). To characterize the relevance of the observed interactions for the activation process, we have investigated their energetic coupling in mutant cycles. Compared to WT, the Ca²⁺ concentration-responses of respective alanine mutants of Phe 518 and Gln 623 are left-shifted, whereas the response of Trp 619 is virtually unchanged (Fig. 1a, Supplementary Figs. 1b and 8c, d). Similarly, the double mutants of residue pairs on α4 and α6 are strongly left-shifted (Supplementary Fig. 8d). The crosstalk between residue pairs is reflected in a weak coupling observed upon the conversion of the EC₅₀ values to energies, emphasizing a functional interaction in the WT protein (Fig. 8f, Supplementary Fig. 8e). Conversely, we wondered to which degree the observed proximity of the polar side chains of His 518 and Gln 623 in the mutant F518H would stabilize the observed conformation. The pronounced coupling energy of 1.4 kJ mol⁻¹ obtained in a double mutant cycle in the background of the strongly left-shifted F518H illustrates a strong interaction between both residues in the active state of the mutant protein (Fig. 8f, Supplementary Fig. 8e). To probe whether this interaction would trap the protein in an intermediate state by preventing a further separation of α-helices 4 and 6 to fully open the subunit cavity, we have determined the structure of the double mutant F518A/Q623A by cryo-EM (Supplementary Fig. 9, Supplementary Table 1). In this double mutant, the truncation of both side chains precludes an equivalent polar interaction as observed in F518H and would thus permit a relaxation of a potentially strained conformation. However, the structure of this double mutant does not resemble the open cavity of nhTMEM16 and instead shows very similar properties as observed in the mutant N562A, with two predominant channel populations (Supplementary Fig. 9). One of these populations is symmetric with both subunits residing in a Ca²⁺-bound inactive state, the other is asymmetric with one subunit showing an inactive and the other an activated state (Supplementary Fig. 10a–c). Their similar size indicates an open probability of about 30%, which is somewhat higher than observed for N562A. The conformation of the activated subunit strongly resembles its equivalent in N562A, except for a small

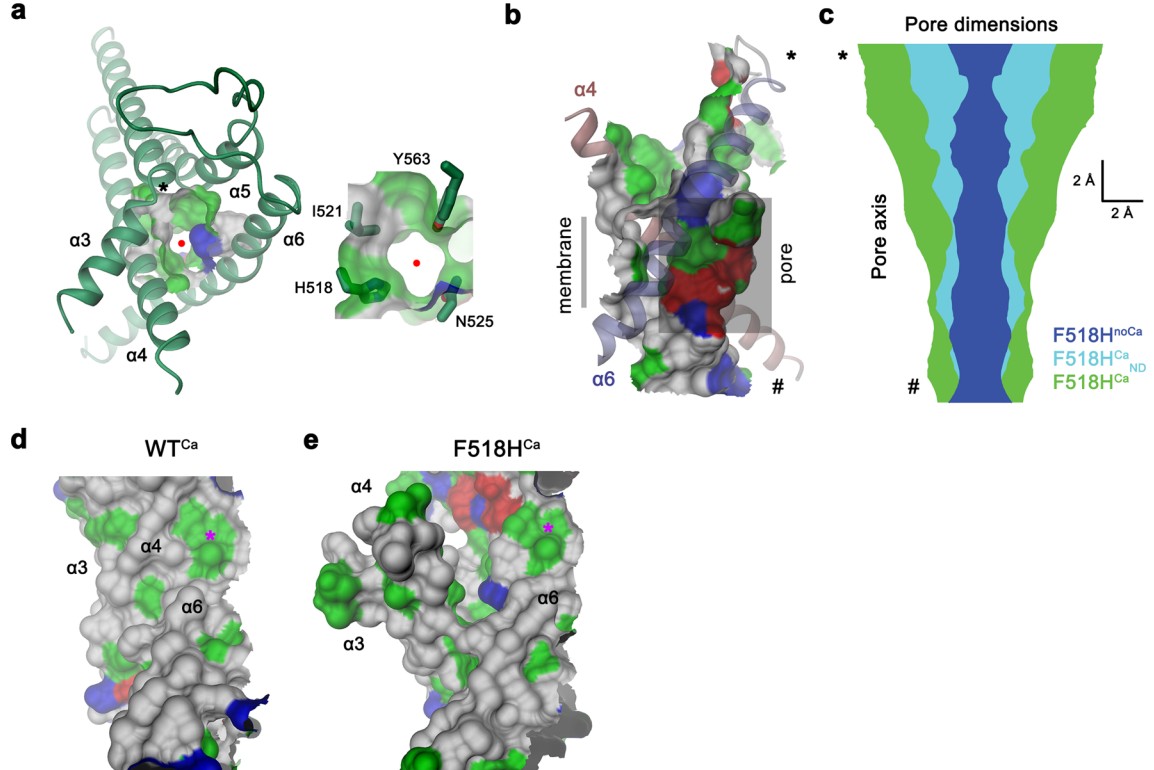

**Fig. 5 | Ion and lipid conduction region. a** Pore region observed in the structure of F518H$^{Ca}$. The view is from the outside. Inset depicts blow-up of the same region with sidechains of selected pore-lining residues shown as sticks. Red circle marks pore center, asterisk, the position of Gly 473. **b** Channel viewed from within the membrane. The protein-enclosed aqueous pore and the molecular surface facing the lipid blayer of the same region are indicated. **c** Pore diameter estimated by Hole[65] mapped along the pore axis from the outside (top) to the inside (bottom) for indicated structures. **b**, **c**, * and # indicate equivalent positions in both panels. The molecular surface of the pore region of WT$^{Ca}$ (**d**) and F518H$^{Ca}$ (**e**) viewed from within the membrane illustrates the exposure of hydrophilic patches to the lipid bilayer in F518H$^{Ca}$ upon activation. Asterisk marks equivalent positions. **a**, **b**, **d**, **e** The molecular surface is colored according to the properties of contacting residues (blue for basic, red for acidic and green for polar residues).

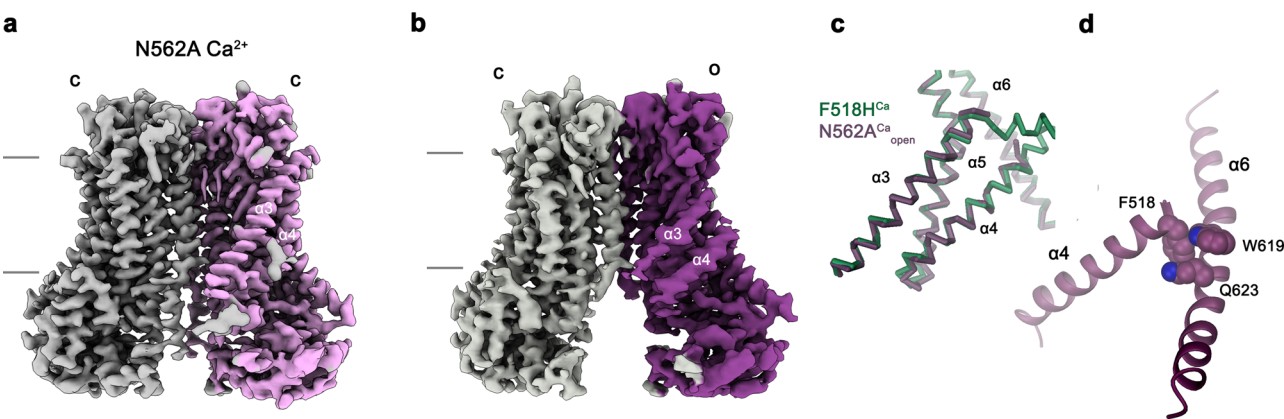

**Fig. 6 | Features of the activating mutant N562A.** Cryo-EM densities of the symmetric closed dimer of N562A$^{Ca}$ at 3.01 Å (**a**) and the asymmetric dimer at 3.49 Å with the activated subunit shown in color (**b**). Subunit conformations are indicated as closed (c) and open (o). **c** Comparison of the open conformation of N562A$^{Ca}$ and F518H$^{Ca}$. Pore-lining helices are shown as Cα-trace. **d** Relationship between α4 and α6 in the activated conformation of N562A$^{Ca}$. Residues that are in contact in the activated subunit are shown as space-filling models.

movement of α4 resulting from the removal of two bulky side chains, and there is no hint of a further opening of the cavity (Supplementary Fig. 10d, e), which provides another piece of evidence for conformational differences in the activated conformations of TMEM16F and nhTMEM16.

Collectively, the results obtained from the rigidification of a conserved glycine residue and the investigation of the new interaction region observed in the Ca$^{2+}$-bound, but not in the Ca$^{2+}$-free state of activating mutants, underline the relevance of the determined structures to describe essential features of TMEM16F activation.

## Discussion
Our study has investigated the conformational transitions promoting ion and lipid conduction in TMEM16F. Previously, the subunit cavity,

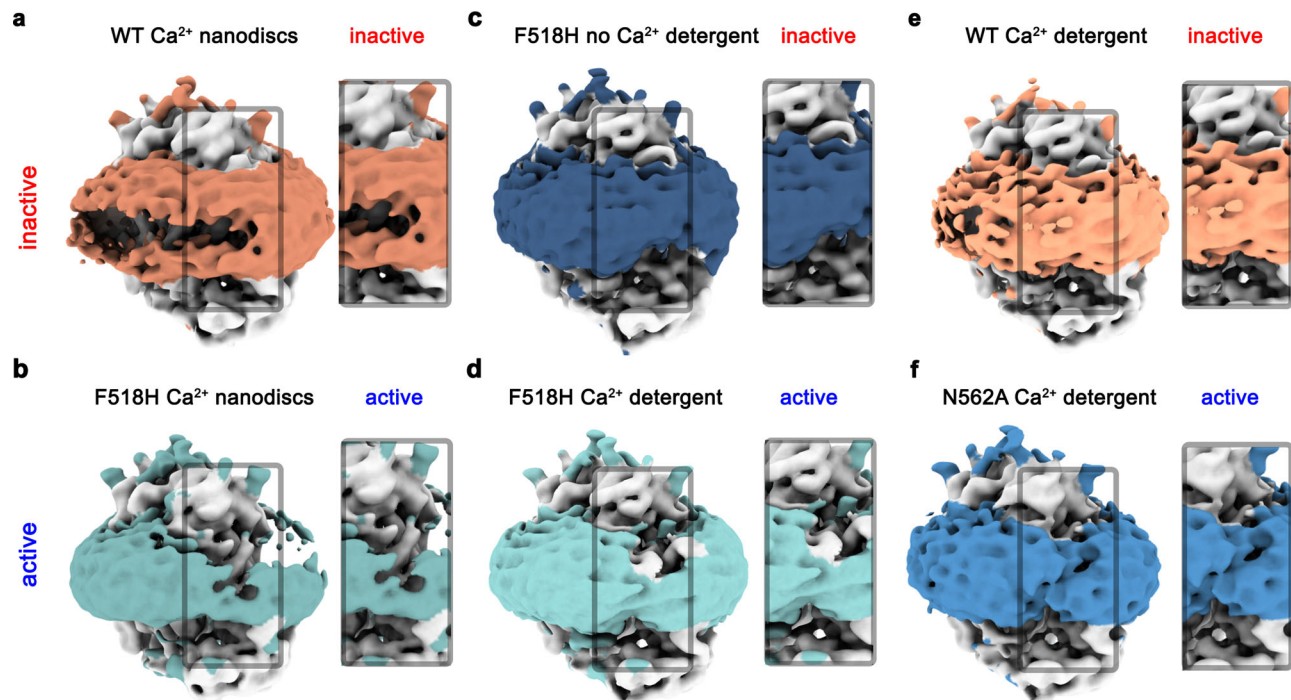

**Fig. 7 | Structural properties of the detergent and lipid region in inactive and active states.** Densities of **a** Ca²⁺-bound TMEM16F WT in 2N2 lipid nanodiscs (PDBID 6QPC, WT^Ca_ND), **b** the Ca²⁺-bound TMEM16F F518H mutant in 2N2 lipid nanodiscs (F518H^Ca_ND), **c** the Ca²⁺-free TMEM16F F518H mutant in digitonin (F518H^noCa), **d** the Ca²⁺-bound TMEM16F F518H mutant in digitonin (F518H^Ca), **e** Ca²⁺-bound TMEM16F WT in digitonin (PDBID 6QP6, WT^Ca), and **f** the Ca²⁺-bound TMEM16F N562A mutant (N562A^Ca) in the asymmetric state viewed towards the activated subunit. All maps were low-pass filtered at 7 Å. The colored region depicts the detergent micelle or nanodisc belt. **a–f** Insets show zoom into highlighted region. The view is towards the subunit cavity of one subunit. The functional state of the respective structures (active, inactive) is indicated.

consisting of α-helices 3–7, was identified as the site of permeation in fungal lipid scramblases and the human protein TMEM16K[18,20,25,29]. The same region constitutes the pore of the anion channel TMEM16A[30,43]. Though not supported by existing structures[19,37], we have assumed by analogy that the corresponding part of TMEM16F would serve a similar role. To investigate the relevance of residues lining the core of the closed subunit cavity for the stability of distinct functional states, we have characterized the properties of point mutants and identified positions whose mutation exerted an equally strong effect on ion and lipid conduction, emphasizing the common origin of their activation mechanism (Figs. 1 and 2). The identified residues conform with a previous study on the localization of a putative gate region in TMEM16F[40] and they are found in proximity to the gate residues identified to stabilize the closed state of the anion channel TMEM16A[31] (Fig. 9a, b).

The structure of activating mutants determined by cryo-EM revealed the desired mechanistic insight into the activation of TMEM16F (Figs. 3, 6 and 8g, h). In our study, the most comprehensive view was provided by three structures of the mutant F518H. In the absence of Ca²⁺, its structure is unaltered compared to the equivalent state of WT, emphasizing that the mutation was well tolerated and does not distort the protein architecture (Fig. 3a–c, Supplementary Fig. 6a). In contrast, the structures of Ca²⁺-bound states in detergent and lipid nanodiscs, reveal extended conformational transitions compared to WT that are pronounced for α-helices 3, 4 and 6 (Fig. 3d–i, Supplementary Fig. 6b–d). While sharing general features, these changes are more extended in the detergent structure F518H^Ca than in the nanodisc structure F518H^Ca_ND, which appears to adopt an intermediate on the transition from F518H^noCa to F518H^Ca presumably defining the activation process (Fig. 4a–c). Although the large 2N2-MSP nanodiscs provide a condition that is closer to a membrane environment than a detergent micelle, the accessible conformational space of the protein might be restricted by the size of the disc and the apparent destabilization of the bilayer at the site of the subunit cavity (Fig. 7b), leading to the observed conformational preference. The conformation of F518H^Ca is stabilized by interactions between the introduced His 518 and residues on α6 (*i.e.* Trp 619 or Gln 623), that are not in contact in the inactive state of F518H^noCa (Fig. 4e, f and 8e, f). Remarkably, we find a similar arrangement in the corresponding structures of the activating mutants N526A^Ca, which concerns a residue that is neither located in the α4-α6 interface nor lining the pore (Fig. 6c, d) and the double mutant F518A_Q623A, where both side chains forming a strong polar interaction in Ca²⁺-bound states of F518H are truncated (Fig. 8f–h). The structural resemblance of three TMEM16F mutants with activating properties underline the notion that the observed conformations carry essential features of an activated state (Fig. 4k, l, 6c, d and 8g, h).

The comparison of TMEM16F structures with known structures of the scramblase nhTMEM16 (ref. 25) and the channel TMEM16A[30,42] sheds light on mechanistic differences underlying the distinct functional properties of the three proteins (Fig. 9a–c). Although promoting passive lipid transport, the sequence relationship of TMEM16F is closer to the anion channel TMEM16A than to nhTMEM16, which is reflected in features of TMEM16F that are not found in other characterized TMEM16 scramblases[19]. A common characteristic of all TMEM16 proteins concerns the reorientation of the intracellular half of α6 in response to Ca²⁺-binding and the consequent coupling of this movement to α4, which trigger either the opening of a protein-enclosed ion conduction pore as it is the case for TMEM16A[30,31,44], or a path for lipid diffusion as found in nhTMEM16 and its relatives[20,25,29]. In the latter proteins, this leads to the lateral movement of α4, which in nhTMEM16 is facilitated by glycine and proline residues that are not conserved in TMEM16A and F (Fig. 9c)[25]. The consequent dissociation of its interaction with α6, results in the exposure of the hydrophilic subunit cavity to the membrane, offering a continuous hydrophilic path for lipid headgroups across the hydrophobic core of the bilayer[25,27]. In combination

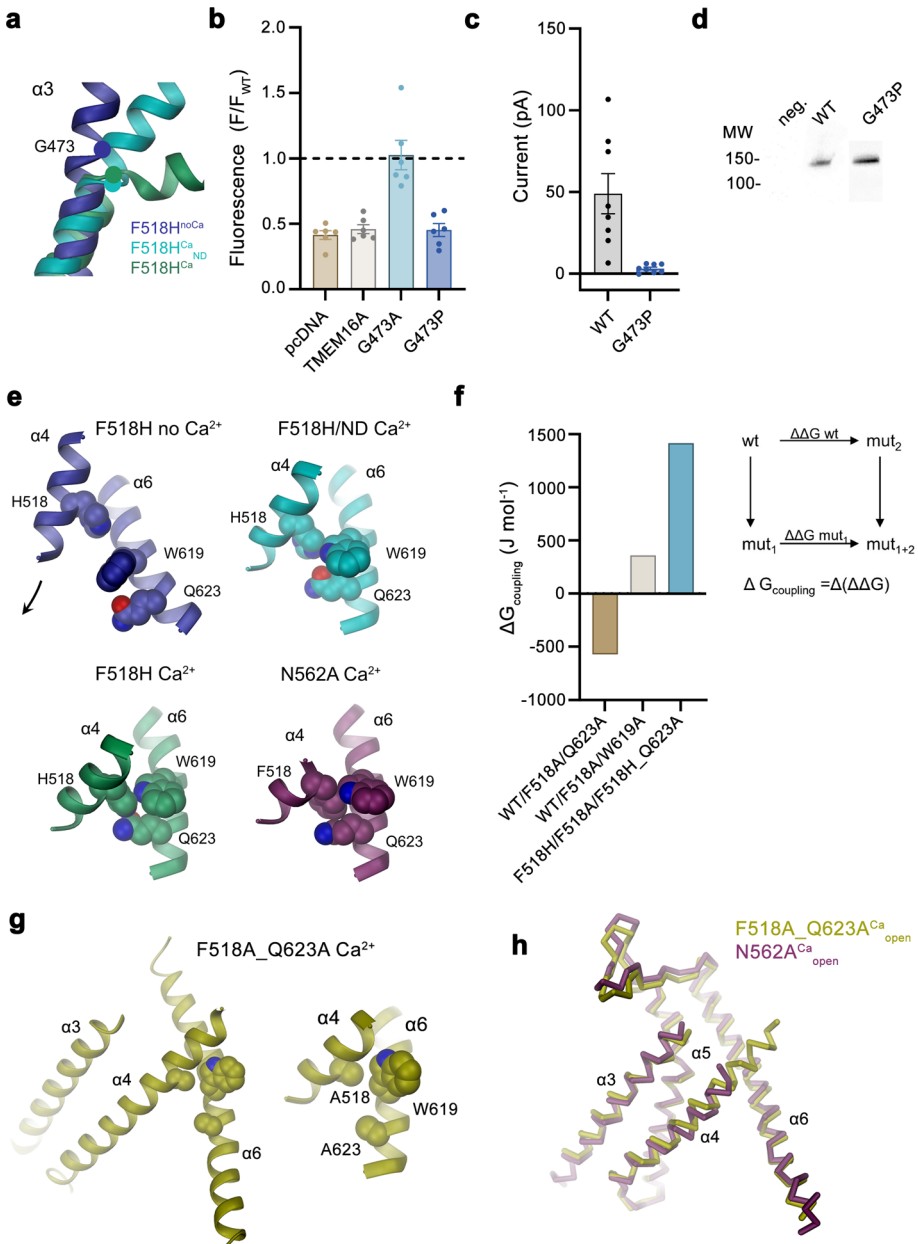

**Fig. 8 | Functional relevance of residues during activation. a** Conformation of α3 around the hinge residue Gly 473 (circle) in structures obtained for the mutant F518H. **b** Lipid transport activity of Gly 473 mutants quantified in a cellular scrambling assay. Data shows fluorescence levels normalized to WT, at elevated $Ca^{2+}$, 600 s after application of ionomycin. Dashed line indicates mean value of WT. Bars show mean of six biological replicates (depicted as spheres), errors are s.e.m. **c** Current magnitudes of HEK293T cells expressing WT or G473P recorded in excised patches. Bars represent mean of individual experiments ($n = 8$), errors are s.e.m. **d** Surface expression of G473P. Anti-myc Western blot corresponds to bio-tinylated constructs pulled-down from HEK293T cells expressing myc-tagged WT or G473P after surface biotinylation. Samples are derived from the same experiment, gels and blots were processed in parallel. **e** Interactions between residues on α-helices 4 and 6 that are in contact in the active but not the inactive state of TMEM16F. **f** Coupling energies between residues depicted in **e**. Inset shows scheme of the double-mutant cycle analysis. WT/F518A/Q623A and WT/F518A/ W619A refer to the cycle quantifying the effect of mutations F518A and either Q623A or W619A relative to WT, F518H/F518A/F518H_Q623A to the cycle of F518A and F518H_Q623A relative to the mutant F518H. **g** Contact region between α-helices 4 and 6 in the activated state of the double-mutant F518A_Q623A$^{Ca}$. Blow up shows residues that are in contact as space-filling models. **h** Comparison of the open conformation of the active subunits of N562A$^{Ca}$ and F518A_Q623A$^{Ca}$. Pore-lining helices are shown as Cα-trace.

with the observed membrane distortion mediated by the protein, this opening is considered a hallmark of a scramblase that facilitates lipid diffusion[22,27]. In contrast, in TMEM16F the conformational change upon $Ca^{2+}$-binding leads to the inward movement of α4, the dissociation of its contacts with α6 and a straightening of its bent conformation assumed in the closed state (Fig. 4a, b). The described transition is accompanied by a concerted relocation of the interacting α3 towards the dimer interface and a pronounced change of its conformation on the extracellular halve (Fig. 4a, c) as illustrated in the comparison of known WT and mutant structures (Supplementary Movie 1). These rearrangements are accompanied by a substantial increase in the exposure of the subunit cavity to the membrane (Fig. 5d, e) and a pronounced distortion of the surrounding lipid and detergent belt that is not found in inactive conformations (Fig. 7). However, in contrast to nhTMEM16 and its relatives, newly established contacts between α4 and α6 prevent a complete opening of the cavity to the membrane and instead lead to the

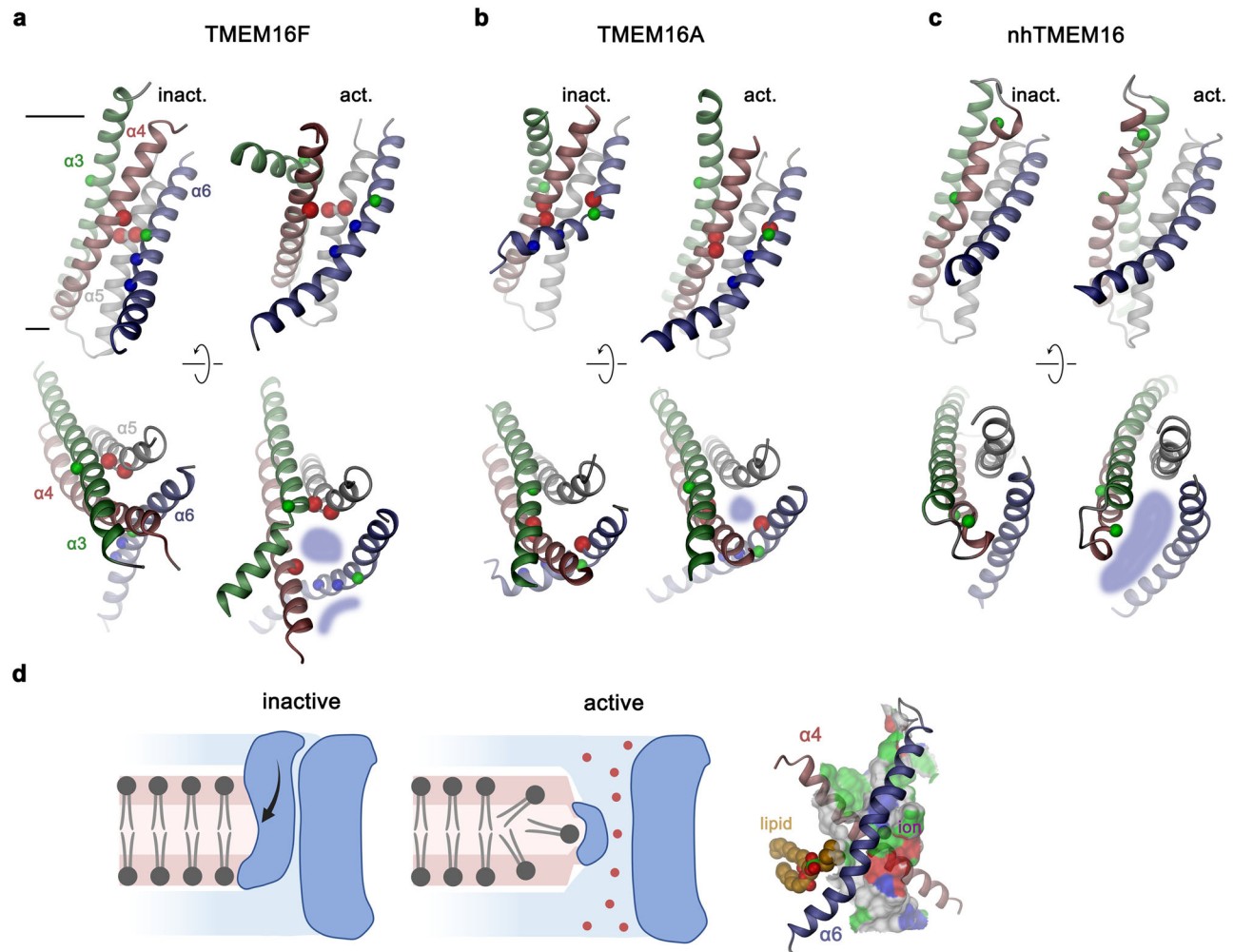

**Fig. 9 | Activation mechanisms.** Conformational differences between putative active and inactive conformations of the pore region in TMEM16F (**a**), TMEM16A (**b**) and nhTMEM16 (**c**) viewed from within the membrane (top) and from the extracellular side (bottom). Lines in (**a**) delineate membrane boundaries. Blue shaded areas indicate the sites of ion and lipid permeation. Helices are shown in unique colors. Spheres highlight Cα positions of selected residues: red, positions whose mutation stabilize the active state; blue, residues on α6 of TMEM16F interacting with α4 in the active state; green, glycine residues that are relevant for conformational changes. **d** Proposed model of ion and lipid permeation in TMEM16F. Left, schematic depiction of the inactive and the active conformation of TMEM16F facilitating ion and lipid conduction in a single conformation. The exposure of polar residues on the common permeation path at the extracellular entrance distorts the bilayer structure thereby facilitating lipid transport. This path diverges in the center of the membrane where ions permeate through an aqueous pore and lipids diffuse on the outside. Right, constricted part of the pore of TMEM16F defined in the structure F518H^Ca viewed from within the membrane. α-helices 4 and 6 forming the interaction region that seals the pore towards the membrane are shown as ribbon. The hypothetical location of a permeating lipid and an ion are indicated. The pore surface is colored according to the properties of contacting residues (blue for basic; red for acidic; green for polar).

formation of a protein-enclosed pore with a constricting diameter of 5 Å bridging a distance of about 12 Å in the center of the bilayer (Fig. 5a–c, Supplementary Fig. 6d). This pore is shielded from the hydrophobic membrane core for about half of its thickness, whereas the contiguous regions towards the extracellular and intracellular side would be accessible to lipids (Figs. 5b, 9d, Supplementary Fig. 6d). The observed conformation bears characteristics of both functional properties of TMEM16F as ion channel and lipid scramblase. Whereas the exposure of hydrophilic surfaces to the membrane, as a consequence of the partial opening of the subunit cavity on its extracellular side, might sufficiently lower the barrier to facilitate the diffusion of lipid headgroups between both leaflets of the membrane, the creation of an aqueous pore would support ion conduction (Fig. 9d). The larger diameter and shorter constricting length of this pore deviates from known TMEM16A structures and could explain the distinct ion conduction properties of TMEM16F, which unlike TMEM16A is non-selective and was even shown to conduct Ca²⁺ (ref. 34, 45, 46). These observations are also consistent

with the effect of the blocker 1PBC, which inhibits TMEM16A and B by binding to a site located at the extracellular entrance of the pore[42]. The same molecule has no effect on TMEM16F, despite the general conservation of interacting residues, emphasizing differences in the activated conformation of TMEM16F compared to anion channels of the family[42].

Collectively, our results have shed light on the unusual functional properties of TMEM16F acting at the same time as ion channel and lipid scramblase. The structures of activating mutants have revealed suitable molecular features to facilitate both functions within a single protein conformation (Fig. 9d). There, the pronounced change in the accessibility of residues lining the subunit cavity exposes hydrophilic residues to the membrane that distort the bilayer, whereas the concomitant opening of an aqueous pore promotes non-selective ion conduction. Although we cannot exclude a further opening of the subunit cavity as mandatory step to adopt a scrambling competent conformation, as suggested by a model of TMEM16F deposited in the

alpha-fold database[47] and a recent study based on computer simulations[48], the membrane distortion observed in our data could be sufficient to facilitate lipid diffusion between leaflets, without requiring continuous protein-lipid interactions in a delimited hydrophilic furrow. A related process was previously proposed in case of fungal scramblases and termed 'out-of-the groove' mechanism[22,39]. Although previous studies have considered the possibility that ion and lipids could diffuse along distinct trajectories in a single protein conformation[19,37], such scenarios remained hypothetical to this point. By contrast, our study reveals an activated conformation of TMEM16F that likely allows simultaneous conduction of ions and lipids. Due to the proximity of their permeation paths, a mutual interaction between both processes appears possible, which could potentially lower the barrier for the diffusion of charged lipids[33,49]. Mechanistic differences between fungal scramblases and TMEM16F could underlie the requirement of a tight regulation for its scrambling activity to prevent activation at resting $Ca^{2+}$-concentration[19], which would be deleterious to the cell. Additionally, the structural differences of its activated state compared to TMEM16A emphasize the necessity of a specific pharmacology for TMEM16F, whose inhibition was proposed as promising strategy to combat viral infections[10].

## Methods
All lipids were obtained from Avanti Polar Lipids, all other consumables from Merck unless stated otherwise.

### Cell culture
TMEM16F KO cells[41] and a murine TMEM16F tetracycline-inducible stable cell line[19] were adapted to suspension culture and grown at 37 °C, 5% $CO_2$ in HyClone HyCell TransFx-H medium (Cytiva) supplemented with 1% fetal bovine serum, 4 mM L-glutamine, 1 g/l poloxamer 188 and 100 U/ml penicillin-streptomycin or alternatively in FUJI BalancED HEK 293 medium supplemented with 1% fetal bovine serum, 4 mM L-glutamine and 100 U/ml penicillin-streptomycin. Adherent cultures of WT HEK293T (ATCC) and TMEM16F KO cells were grown in high glucose DMEM medium supplemented with 10% fetal bovine serum, 2 mM L-glutamine, 1 mM sodium pyruvate and 100 U/ml penicillin-streptomycin at 37 °C and 5% $CO_2$. Cells were routinely tested for mycoplasma contamination.

### Construct preparation
Murine TMEM16F constructs were prepared as described[19]. Briefly, the sequence of TMEM16F was cloned into FX-compatible vectors[50] and point mutations were introduced with a modified QuikChange protocol[51] (Supplementary Table 2). Constructs used for protein purification and in cellular scrambling assays contain a fusion of a 3 C recognition site, Myc and SBP tags to their C-terminus. Constructs used in electrophysiology experiments contained either a YFP or GFP tag between the 3 C site and the Myc-tag. All TMEM16F constructs were verified by sequencing. The plasmid used for the expression of the membrane scaffold protein (MSP) 2N2 was obtained from Addgene (plasmid #29520). The plasmid used for the expression of eYFP H148Q/I152L (HQ-YPF) was obtained from Addgene (plasmid #25872).

### Protein expression
The over-expression of constructs destined for protein purification was carried-out in suspension culture. WT TMEM16F was expressed in a tetracycline-inducible cell line as described[19], TMEM16F mutant constructs were expressed by transient transfection of TMEM16F KO cells with PEI MAX (Polysciences) at 2.5 times the amount of DNA. After induction or transfection, 3.5 mM valproic acid was added to the cell cultures. Cells were harvested between 35 to 48 h post transfection, washed with PBS, flash-frozen in liquid nitrogen and stored at −80 °C until further use. For the cellular scrambling assay, adherently-cultured HEK293T TMEM16F KO cells were used. Black glass-bottom 96-well plates were coated with poly-L lysine, and cells were seeded at a density of 30–40% 5 h prior to transfection. For transfection, TMEM16F mutant DNA and HQ-YPF DNA were mixed at a ratio of 3:2 and cells were transfected with 100 ng DNA/well, using either Lipofectamine 2000 or 3000 (Thermo Fischer) following the manufacturer's guidelines. For electrophysiology experiments, HEK293T cells were used. Transient transfection was performed with Lipofectamine 2000 at a ratio of 2.5 times the amount of DNA. Cells were used within 24 to 48 h after transfection.

### Cellular lipid scrambling assay
The assay was performed 24 h after transfection of TMEM16F KO cells with the indicated plasmid. For all measurements, the medium was removed and replaced by 50 µl Buffer A containing 10 mM HEPES pH 7.4, 25 mM glucose, 2 mM glutamax, 1.5 mM Na-pyruvate, 140 mM NaCl, 2.5 mM $CaCl_2$, 5% annexin V Alexa Fluor 594 conjugate (Thermo Fischer), and 5 nM SytoxRed (Thermo Fischer). Cells were rapidly transferred to a GE INCellAnalyzer 2500HS microscope (GE). The cells were imaged as a time series at 10x magnification in the green (HQ-YFP), red (Annexin V) and far red (Sytox) channels. Images were acquired every 10 s. After 3 min, 50 µl of Buffer B containing 10 mM HEPES pH 7.4, 155 mM $NaNO_3$, 2.5 mM $CaCl_2$, 5% annexin V Alexa Fluor 594 conjugate, 5 nM Sytox red, and 100 µM ionomycin (Thermo Fischer) were added rapidly and cells were imaged for another 10 min. Data was evaluated in FIJI[52]. Cells exhibiting a YFP signal were selected as ROIs from the green channel and from these apoptotic cells exhibiting signal in the far-red channel were removed. Image Stacks from the red channel were aligned using template matching and the background was subtracted. Fluorescence intensity of the ROIs was quantified for all frames in the red channel. Data analysis was performed in Microsoft Excel. The data is presented as annexin V fluorescence normalized to the WT TMEM16F signal.

### Surface biotinylation
Adherent HEK293T cells at 60% confluency were transfected with 10 µg of plasmid encoding TMEM16F WT and mutant constructs containing a c-terminal SBP-Myc-Venus tag per 10 cm culture dish. Transfection was performed using PEI MAX at a DNA:PEI ratio of 1:4. Surface biotinylation was performed using the Pierce™ Cell Surface Biotinylation and Isolation Kit (Themo Fisher) following manufacturer's guidelines. After washing with PBS, cells from 20 ml culture per sample were biotinylated after 24 h of expression using EZ-Link Sulfo-NHS-SS-Biotin dissolved in 8 ml PBS at a concentration of 0.125 mg/ml for 30 min at 4 °C, unless stated otherwise. For both the WT and the mutant G473P, an equivalent number of cells was incubated with PBS without Biotin as a negative control. The reaction was quenched by addition of 400 µl quenching buffer and cells were harvested. Lysis and purification steps were carried out at 4 °C. After washing with PBS, cells were lysed in buffer containing 20 mM HEPES pH 7.4, 150 mM NaCl, 2 mM EGTA, 2% glycoldiosgenin GDN (Anatrace) and protease inhibitors (cOmplete, Roche) by gentle agitation for 1 h at 4 °C. Insoluble fractions were removed by centrifugation at 20,000 g for 20 min and supernatant was incubated with 400 µl Neutravidin agarose slurry pre-equilibrated in wash buffer (20 mM HEPES pH 7.5, 150 mM NaCl, 2 mM EGTA, 0.03% GDN) for 1 h at room temperature (RT) with gentle mixing of the sample. The flowthrough was discarded and the resin was washed 3 times with 400 µl of wash buffer. The bound biotinylated proteins were eluted by incubation with 400 µl elution buffer (50 mM fresh DTT in Laemmli buffer) for 1 h at RT with gentle mixing of the sample. 20 µl of eluates were loaded and separated by SDS-PAGE. Proteins were transferred onto a PVDF membrane and analysed by Western blot using a mouse-anti-myc primary antibody (Sigma M4439) and HRP-coupled goat-anti-mouse secondary antibody (Jackson ImmunoResearch). Both antibodies were dissolved 1:10000 in TBS-T with 5% skimmed milk. The chemoluminescence signal was developed using ECL prime reagent (Cytiva) and recorded with a Fusion FX7 imaging system (Vilber).

## Electrophysiology

Transfected cells were identified by YFP fluorescence and used for patch-clamp experiments within 24–30 h post transfection. All recordings were performed in the inside-out configuration[53] at 20 °C. Patch pipettes were pulled from borosilicate glass capillaries (OD 1.5 mm, ID 0.86 mm) and were fire-polished using a microforge (Narishige). Pipette resistance was typically 3–5 MΩ when filled with pipette solutions. Inside-out patches were excised from HEK293T cells expressing the TMEM16F construct of interest after the formation of a tight seal. Seal resistance was typically 4–8 GΩ. Voltage-clamp recordings were performed using an Axopatch 200B (Molecular devices) amplifier controlled by the Clampex 10.7 software (Molecular Devices) through Digidata 1440 (Molecular Devices) and analyzed with Clampfit 10.7 (Molecular Devices). The data were sampled at 10 kHz and filtered at 1 kHz. Solution exchange was achieved using a double-barreled theta glass pipette mounted on an ultra-high speed piezo-driven stepper (Siskiyou). Liquid junction potential was not corrected. All recordings were measured in symmetrical NaCl solutions. Step-like solution exchange was elicited by analogue voltage signals delivered through Digidata 1440. Steady-state currents were measured for each $Ca^{2+}$-concentration. The background current was recorded in $Ca^{2+}$-free solution and subtracted prior to analysis. The $Ca^{2+}$-free solution was used as the pipette solution. $Ca^{2+}$-free intracellular solution contained 150 mM NaCl, 5 mM EGTA, and 10 mM HEPES, pH 7.40. High intracellular $Ca^{2+}$ solution, with a free concentration of 1 mM, contained 150 mM NaCl, 5.99 mM $Ca(OH)_2$, 5 mM EGTA, and 10 mM HEPES, pH 7.40. The pH was adjusted using 1 M NMDG-OH solution. Intermediate $Ca^{2+}$ solutions were obtained by mixing high $Ca^{2+}$ and $Ca^{2+}$-free intracellular solutions at the ratio calculated according to the WEBMAXC calculator[54]. To correct for the rundown of TMEM16F current in excised patches, we used a previously described method[23] where the test current magnitude is normalized to the average magnitude of a pre-and post-reference $Ca^{2+}$-pulse at which TMEM16F exhibits full current. The concentration–response data was normalized to the lowest and highest mean value in each dataset and fitted to the Hill equation using GraphPad Prism 9. A separate set of data was recorded for experiments on mutant cycles displayed in Fig. 8f and Supplementary Fig. 8c–e, using the same set of buffers to avoid small fluctuations in the $Ca^{2+}$ concentration and increase the precision of measurements. The calculation of coupling energies was based on the ratio of $EC_{50}$ values, which assuming a Del-Castillo Katz mechanism, would reflect the ratio of forward equilibrium constants in case of unaffected $K_d$ values of $Ca^{2+}$ binding to the closed state of the protein and high efficacies in a $Ca^{2+}$-bound state. $EC_{50}$ values for mutant cycles were either obtained by an unconstrained fit or a fit where the Hill-coefficient was constrained to a value of 2, which corresponds to previously determined values for TMEM16F[19] and TMEM16A[23]. Results of respective fits are displayed in Fig. 8f and Supplementary Fig 8e.

## Protein purification

Protein purification was performed at 4 °C and completed within 10 h. Cell pellets were solubilized in 20 mM HEPES pH 7.5, 150 mM NaCl, 5 mM EGTA and 2% Digitonin (Applichem) for most constructs and 2% GDN (Anatrace) for F518A_Q623A under gentle agitation for 2 h. Lysate was cleared by ultracentrifugation at 85,000 g for 30 min. Supernatant was bound in batch to Strep-tactin resin (IBA Lifesciences) pre-equilibrated with SEC buffer (20 mM HEPES pH 7.4, 150 mM NaCl, 2 mM EGTA, 0.1% Digitonin (Millipore) for most constructs and 0.03% GDN for F518A_Q623A) for 2 h under gentle agitation. Resin was washed with 60 column volumes (CV) of SEC buffer and the target protein was eluted with 6 CV SEC buffer supplemented with 5 mM d-Desthiobiotin. Eluate was concentrated and loaded onto a Superose 6 10/300 GL column (Cytiva) pre-equilibrated with SEC buffer. Peak fractions were collected and concentrated to 2 mg/ml.

Nanodisc reconstitution was performed as described previously[19] with modifications detailed below. The lipid composition chosen for nanodisc reconstitution was 3 POPC:1 POPG. Chloroform-dissolved lipids were initially dried under a nitrogen stream, washed with diethyl ether and dried again under a nitrogen stream and by vacuum desiccation overnight. The lipid mix was solubilized to a final lipid concentration of 10 mM in 30 mM DDM (Anatrace). Initial steps in the protein purification proceeded as described above. Prior to the elution with desthiobiotin from Strep-tactin resin, the described lipid mixture was added and incubated with the immobilized protein for 40 min under gentle agitation. Subsequently, MSP 2N2 was added and the sample was incubated for extra 40 min. Nanodiscs were assembled at a final molar ratio of 1:220 of MSP 2N2:lipids. 50 mg SMII Biobeads (BioRad) per mg of detergent were added for removal of DDM and Digitonin and the sample was kept under gentle agitation overnight. Empty nanodiscs were removed by washing the resin with 60 CV of SEC buffer without detergent. Nanodiscs containing protein were eluted with 10 CV of SEC buffer without detergent supplemented with 5 mM d-Desthiobiotin. Eluate was concentrated and loaded onto a Superose 6 10/300 GL column pre-equilibrated with SEC buffer without detergent. Peak fractions were collected and concentrated to 1 mg/ml.

## Protein reconstitution and liposome-based scrambling assay

Reconstitution and liposome-based scrambling assays were performed as described previously[19]. In brief, purified WT and mutant TMEM16F were reconstituted into liposomes containing trace amounts (0.5% w/w) of fluorescently (nitrobenzoxadiazole, NBD) labeled lipids. As lipid mix, soybean polar lipids extract supplemented with 20% cholesterol (mol/mol) and 0.5% 18:1–06:0 NBD-PE was used. Liposomes were prepared as previously described[55]. Dry lipid films were solubilized in 20 mM HEPES pH 7.5, 300 mM KCl and 2 mM EGTA (buffer A) in a final concentration of 20 mg/ml, sonicated, subjected to three freeze-thaw cycles in liquid $N_2$ and stored at −80 °C. Immediately prior to reconstitution, the lipids were extruded 21 times through a 200 nm pore polycarbonate membrane and diluted to 4 mg/ml in buffer A. Liposomes were destabilized by addition of Triton X-100 aliquots in 0.02% steps until the onset of lipid solubilization monitored by the decrease of the scattering at 540 nm in a spectrophotometer. After addition of another 4 aliquots of 0.02% Triton X-100, the protein was added at a lipid-to-protein ratio of 100:1 (w/w) to the destabilized liposomes and the mixture was incubated at RT for 15 min under gentle rotation. Subsequently, 20 mg of SMII biobeads per mg of lipids were added four times to completely remove digitonin and Triton X-100. The sample was moved to 4 °C after 45 min. Proteoliposomes were harvested after 24 h by filtration to remove the biobeads and pelleted by centrifugation at 150,000 g for 30 min. The proteoliposomes were resuspended in buffer A with solutions containing the appropriate amount of $Ca(NO_3)_2$ to obtain the indicated concentrations of free $Ca^{2+}$ calculated with the online WEBMAXC calculator[54]. The liposomes were resuspended at 10 mg/ml and subjected to three freeze-thaw cycles in liquid $N_2$ and stored at −80 °C until further use.

The liposome scrambling assay was carried out at RT. Thawed proteoliposomes were extruded through a 200 nm membrane pre-equilibrated in buffer A containing the desired amount of free $Ca^{2+}$. For each measurement, proteoliposomes were diluted to 0.2 mg/ml in 80 mM HEPES pH 7.5, 300 mM KCl, 2 mM EGTA and $Ca(NO_3)_2$ (buffer B). Scrambling was assayed essentially as described for other scramblases of the TMEM16 family[18,35] in symmetric buffer conditions by quantifying NBD fluorescence with an excitation wavelength of 470 nm and emission wavelength of 530 nm in a spectrofluorometer for a total recording time of 400 s. After 60 s of recording, 30 mM sodium dithionite was added to irreversibly bleach exposed NBD groups. The sample was constantly stirred during the measurement. The NBD-fluorescence decay was plotted as $F/F_{max}$. In our experiments, we found measurements between different sets of proteoliposomes to be most suitable for comparison[19], if the proteins were reconstituted at the same time with the same batch of destabilized liposomes. To ensure optimal

comparability between the samples, WT TMEM16F was thus always purified and reconstituted along with the mutants and any data obtained was only compared to data from the same reconstitution. The $Ca^{2+}$-dependence of the scrambling activity was quantified by measuring the NBD-fluorescence at the respective $Ca^{2+}$-concentration 140 s after addition of dithionite and expressed as $1-F^{Ca^{2+}}/F_{WT}^{0Ca^{2+}}$, as there is no appreciable scrambling activity for wt TMEM16F in the absence of $Ca^{2+}$. The data was normalized to the value of WT at 1 mM $Ca^{2+}$ and fitted to a Hill Equation. Protein reconstitution of all investigated constructs was assayed by solubilization proteoliposomes in 5% GDN and quantification of the tryptophan-fluorescence on FSEC using a Superose 6 5/150 GL column equilibrated with SEC buffer. In parallel, the reconstitution efficiency of WT and F518H was compared by Western-blotting using an anti-myc antibody as described above.

## Electron microscopy sample preparation and data collection

Immediately after purification, 2.5 μl of protein was applied on cryo-EM grids, blotted for 2–5 s in a Vitrobot (Mark IV, Thermo Fisher) at 4 °C temperature and 100% humidity, and subsequently plunge-frozen in liquid ethane/propane mix. Digitonin-solubilized protein was frozen at a concentration of 1.5–2 mg/ml on holey-carbon grids (Quantifoil Au R1.2/1.3, 200 mesh), whereas protein reconstituted into nanodiscs was frozen at a concentration of 1 mg/ml on holey gold grids (UltraAUfoil, 300 mesh) and at 0.2 mg/ml on graphene-oxide covered copper grids (EMS Scientific, 400 mesh) due to the expected preferred orientation of the sample. All grids were glow-discharged at 5 mA for 30 s prior to sample application. In case of the datasets obtained for $Ca^{2+}$-bound structures, samples were supplemented with 2 mM free $CaCl_2$ and, in case of detergent-solubilized F518H and N562A mutant, additional 0.01 mM diC8-$PI_{(4,5)}P_2$ ($PIP_2$) shortly before freezing.

Grids were stored in liquid $N_2$ until further use. For all datasets, cryo-EM data was collected on a 300 keV Titan Krios Microscope (Thermo Fisher) using a post-column energy filter (Gatan) in zero-loss mode, a 20 eV slit, a 100 mm objective aperture, in an automated fashion using EPU software (Thermo Fisher) on a K3 detector (Gatan) in counting mode. Cryo-EM images were acquired at a magnification of 130,000x with a defocus range of −1 to −2.4 μm in super-resolution mode at a calibrated pixel size of 0.3255 Å and binned to 0.651 Å during data acquisition. The exposure time per movie was 1.26 s and the total electron dose tuned to 64.2 $e^-$/Å². Cryo-EM data collection for one of the three datasets of the nanodisc sample was performed with a stage tilt of 20°. The stage was tilted after alignment of the microscope and immediately before the start of data collection.

## Image processing

All datasets were processed using cryoSPARC v.2.7–3.1 (ref. [56]). Dose-fractionated movies were imported and processed using the patch motion correction and patch CTF estimation tools from cryoSPARC. Movies showing excessive contaminations, high total full-frame motion and low-resolution CTF estimation were discarded at this stage. Image processing was performed with some adaptations between different datasets. For the F518H$^{Ca}$ dataset, templates were generated from a subset of 60 micrographs using the blob picker tool and subsequent 2D classification. Three templates sampling the x- y- and z- view on the protein were lowpass-filtered to 20 Å and used for subsequent template picking on the entire dataset. For all datasets, particles were extracted with a box size of 520 Å and binned during extraction by ½ to a final pixel size of 1.302 Å. After several rounds of 2D classification with gradually increasing initial uncertainty factors, an ab initio model was generated from the best classes. The clean particle set was subjected to heterogeneous refinement using the best ab initio class and a decoy 3D class generated from decoy 2D classes of the same dataset. The particles sorted into the best class were refined using non-uniform refinement[57] with imposed C2 symmetry. In case of the F518H$^{noCa}$ dataset, low-pass filtered templates obtained for the $Ca^{2+}$-

bound dataset were used for template picking. For 2D classification and 3D refinement, particles were treated similarly to the F518H$^{Ca}$ dataset. After non-uniform refinement, the particles were subjected to local and global CTF refinement. For the N562A dataset, low-pass filtered (20 Å) back projections of the F518H $Ca^{2+}$-free structure were used for template picking. The dataset was treated similarly, except that non-uniform refinement was performed in C1 symmetry, since asymmetry between the two subunits of the TMEM16F dimer was observed. To better separate the different conformational states, 3D variability analysis[58] of the dimer was performed and the clean particles were subjected to heterogeneous refinement using the 10 Å low-pass filtered start and end frame of the 3D variability analysis result as input volumes. The resulting maps, depicting open/closed (o/c) and closed/closed (c/c) states of subunits were subsequently separately refined using non-uniform refinement with C1 symmetry for the asymmetric and C2 symmetry for the symmetric class.

In the case of the F518H$^{Ca}_{ND}$ datasets, preliminary templates were generated with the blob picker tool as described for the F518H$^{Ca}$ dataset and used to train a Topaz model[59] for improved particle picking. A model was trained for each of the three recorded datasets. Particles from all three datasets were polished by separate rounds of 2D classification and eventually combined for a final round of 2D classification. As the data suffered from severe preferential orientation, oversampled views were removed using the rebalance 2D tool from cryoSPARC. The resulting particles were then used to create an ab intio 3D model with 2 classes. The particles were heterogeneously refined using the two classes from the ab initio model as input volumes. The better class was subjected to non-uniform refinement in C1 symmetry and local refinement. 3D variability analysis was used to deduct different conformational states in analogy to the processing of N562A in detergent. For the F518A_Q623A$^{Ca}$ dataset, preliminary templates were generated from a subset of micrographs with the blob picker tool as described for the F518H$^{Ca}_{ND}$ dataset and used to train a Topaz model[59]. The picked particles were cleaned in 2 subsequent rounds of 2D classification. As, among collected datasets, the F518A_Q623A$^{Ca}$ dataset displayed the most severe preferential orientation, the ab-initio model of the N562A$^{Ca}$ mutant was low-pass filtered to 20 Å and provided as a 3D reference for heterogenous refinement of the particles with a decoy class as described above. The oversampled views were removed using the rebalance 2D tool from cryoSPARC. To separate the open/closed (o/c) and closed/closed (c/c) conformational states, the particles were treated similarly as described for the N562A mutant, using 3D variability analysis and heterogenous refinement onto the low-pass filtered end-frames. Due to the preferential orientation of the particles, both F518A_Q623A maps display a pronounced anisotropy. In all cases, local resolution was computed using cryoSPARC[56].

## Model building and refinement

All models were built in Coot[60] using full-length structures of TMEM16F as reference (i.e. PDBID 6QP6 for structures obtained in presence and 6QPB for the structure in absence of $Ca^{2+}$). The atomic models were improved iteratively by cycles of real-space refinement in PHENIX[61] with secondary structure constraints applied followed by manual corrections in Coot. Validation of the models was performed in PHENIX. Surfaces were calculated with MSMS[62]. Figures and movies containing molecular structures and densities were prepared with DINO (http://www.dino3d.org), UCSF Chimera 1.15 (ref. [63]), and ChimeraX 1.2.5 (ref. [64]). Pore diameters were calculated with Hole (ref. [65]).

## Statistics and reproducibility

Western Blots shown in Fig. 8d and Supplementary Fig. 2a were performed once. Electron Micrographs shown in Supplementary Fig. 3a, 4a, 5a, b, 7a, 9a are representative for the entire dataset (Number of

images F518H$^{noCa}$, $n = 18,850$, F518H$^{Ca}$, $n = 13,235$, F518H$^{Ca}_{ND}$, $n = 5,792$ (a) and $n = 7,325$ (b), N562A$^{Ca}$, $n = 9,088$, F518A_Q623A$^{Ca}$, $n = 10,482$). Addition of further data for electrophysiology and scrambling experiments did not alter the displayed results.

## Reporting summary

Further information on research design is available in the Nature Portfolio Reporting Summary linked to this article.

## Data availability

Data supporting the findings of this study are available from the corresponding authors upon reasonable request. Source data are provided with this paper. The cryo-EM maps, half-maps, and masks have been deposited in the Electron Microscopy Data Bank under accession numbers EMD-15913 (F518H$^{noCa}$), EMD-15914 (F518H$^{Ca}$), EMD-15919 (F518H$^{Ca}_{ND}$), EMD-15917 (N562A$^{Ca}_{OC}$), EMD-15916 (N562A$^{Ca}_{CC}$), EMD-15958 (F518A_Q623A$^{Ca}_{OC}$), EMD-15959 (F518A_Q623A$^{Ca}_{CC}$). Coordinates for models are available in the Protein Data Bank under PDBIDs 8B8G (F518H$^{noCa}$), 8B8J (F518H$^{Ca}$), 8B8Q (F518H$^{Ca}_{ND}$), 8B8M (N562A$^{Ca}_{OC}$), 8B8K (N562A$^{Ca}_{CC}$), 8BC0 (F518A_Q623A$^{Ca}_{OC}$), 8BC1 (F518A_Q623A$^{Ca}_{CC}$). Source data are provided with this paper.

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

## Acknowledgements

This work was supported by a grant of the European Research Council (ERC no 339116, AnoBest) to R. D. The Center for Microscopy and Image Analysis (ZMB) of the University of Zurich is acknowledged for the support and access to the electron microscope. The cryo-electron microscope and K3 camera were acquired with the support of the Baugarten and Schwyzer-Winiker foundations and a Requip grant of the Swiss National Science Foundation. We thank Sonja Rutz and Marta Sawicka for help during cryo-EM data collection and Dawid Deneka for his help in establishing the cellular scrambling assay. Valeria Kalienkova is acknowledged for her advice regarding column nanodisc reconstitution. All members of the Dutzler laboratory are acknowledged for their help at various stages of the project.

## Author contributions

C.A. has carried out the initial screen of TMEM16F mutants, prepared expression constructs, liposome-based scrambling assays and initial cryo-EM studies. V.C.M. and C.P. have supported initial cryo-EM studies. M.A. has prepared samples for cryo-EM, proceeded with data collection and structure determination and carried out structure-based functional experiments and double mutant cycle analysis. M.S.S. has assisted with grid freezing, cryo-EM data collection and data analysis. R.D. and M.A. prepared initial draft of the manuscript. All authors contributed to the final manuscript.

## Competing interests

The authors declare no competing interests.
