## [Peer Review File · Nature Communications]

Structural basis for the activation of the lipid scramblase TMEM16FREVIEWER COMMENTS

Reviewer #1 (Remarks to the Author):

In this manuscript, Arndt et al performed mutation analysis of the phospholipid scramblase TMEM16F, focusing on the proposed catalysis domain. The authors identified several mutations, making TMEM16F more active. By completing the Cryo-EM structural analysis of the particular mutant in Ca²⁺-bound and -unbound states, the authors also revealed the structural alternations occurring at the transmembrane helices, especially at α 3 and α 6. From screening for identification of the particular mutations activating TMEM16F to revealing their effect on the protein structure, this study comprehensively analyzed the activation mechanisms of TMEM16F and obtained mechanistic insight into how TMEM16F undergoes a structural change to scramble lipids. However, before publication, several points need to be clarified.

Major points

1. In Fig 1a, fluorescence level was compared between wild-type and mutants in resting state. However, from this data, you cannot judge whether TMEM16F mutants (such as F518A) are activated like a Ca²⁺-stimulated state. Reference fluorescence ratio of stimulation/resting should be indicated.
2. In Fig 2e and f, fluorescence level was compared between wild-type and mutants in resting state and activated state. Similar to Fig 1a, the Reference fluorescence ratio of stimulation/resting should be indicated.
3. In Fig 2g, when the authors compare WT to F518H in the reconstituted system, how can the authors determine the reconstitution proportion between these two? If the authors argue that F518H is activated more than WT in 1 mM Ca²⁺-stimulated state, the authors also conclude that differences can be observed even in 0 mM Ca²⁺ state?
4. In Fig 3a-c, when the authors compare F518H (Ca²⁺(-)) to WT (Ca²⁺(-)), how about the structural change?
5. In Fig 4a, how the authors can conclude that Nanodisc-reconstituted F518H is an intermediate state? The lipid environment in Nanodisc can be considered a more physiological environment than detergent.
6. In Fig 5d, WB should be shown with a size marker with molecular weight.

Minor points

Consistency on the description in Figures. Ca or Ca²⁺, F518H or N562Ca

Reviewer #2 (Remarks to the Author):

The TMEM16 proteins function either as ion channels or lipid scramblase. Among its family members, THEM16F is unique as it can play both roles in conducting ions across the membrane and moving lipid molecules between the two membrane leaflets. Despite extensive studies, including several structures, how the protein performs both roles remains unclear. In the current work, Arndt and colleagues, combining elegant functional assays and structure determination by cryo-electron microscopy, have provided an answer. Using a cellular lipid transport assay and electrophysiology measurements, the authors found that the same mutations on Helices 4 and 5 affect the activation of both lipid scrambling and ion conductance. Using cryo-EM, the investigators further determined the structures of two of such mutants in the absence and presence of Ca⁺⁺. Comparison of the structures with and without Ca⁺⁺ revealed conformational changes in Helices 4 and 5 that open the ion conductance pore and allow lipid scrambling. Thus, the work has solved a long-standing mystery in the TMEM16 field. The data quality is excellent, and the manuscript is clearly written. The work should certainly be seen by the broad readership of Nature Communications.

One issue that is not discussed, which may be beyond the scope of the current manuscript, is the relationship between ion transport and lipid transport. As both are activated through the same Ca⁺⁺-binding events, is it possible that the two transport processes are coupled? Even a loose coupling mechanism may help lower the energy barrier for transporting charged lipids across the membrane.

1. Page 6, Line 98. To help a reader unfamiliar with the TMEM16 family, the authors can include an overall structure of TMEM16F in Fig. 1 with the parts of interest highlighted.
2. Page 9, Line 176. F516H(noCa) should be F518H(noCa).
3. A scale bar should be included in the electron micrographs, ED Figs. 3a, 4a, 5a, 5b and 7a.
4. In ED Fig. 3c, 18850 movies should be 18,850 movies.
5. Coloring scheme used in local resolution maps in ED Figs. 3f, 4f, 5g, 6f and 6i is a bit confusing. The color scale should be reversed, with the high resolution (3 Å) areas colored blue and the low resolution (5 Å) red, making it similar to the convention for coloring the B-factor.
6. In ED Table 1, the number of movies in each dataset should be included.

Reviewer #3 (Remarks to the Author):

This is a remarkable paper offering new insights that illuminate the curious functional differences between TMEM16F, the first scramblase of the TMEM16 family to be identified, and its fungal scramblase counterparts as well as strict ion channel members of the family. The results are based on mutations, tested in cell-free assays, complemented by several cryo-EM structures in detergent and nanodiscs of select mutants. I would avoid the use of the word catalysis as used in the abstract (line 19) and elsewhere as this has a particular connotation that is not appropriate here. I have no major comments except for a clarification (see below).

Figure 1 and associated extended figure 1a are a little confusing. Annexin binding at rest shows that certain mutants have elevated PS exposure – it might be expected that this effect between the different mutants should be damped out on ionomycin treatment where Ca is presumably in excess. Yet F518 stands out under both conditions.

We thank the reviewers for their kind remarks and constructive comments, which we have considered in our revised manuscript and addressed in detail below. Our revision also contains a novel structure of the double mutant F518A_Q623A. This construct removes a strong interaction between α -helices 4 and 6 found in F518H^{Ca}, which could potentially stabilize the protein in an intermediate state towards activation. However, the structure of F518A_Q623A^{Ca} closely resembles the Ca²⁺-bound conformations of N562A and F518H, thus providing further evidence that all structures represent an activated conformation of TMEM16F.

Reviewer #1 (Remarks to the Author):

In this manuscript, Arndt et al performed mutation analysis of the phospholipid scramblase TMEM16F, focusing on the proposed catalysis domain. The authors identified several mutations, making TMEM16F more active. By completing the Cryo-EM structural analysis of the particular mutant in Ca²⁺-bound and -unbound states, the authors also revealed the structural alternations occurring at the transmembrane helices, especially at α 3 and α 6. From screening for identification of the particular mutations activating TMEM16F to revealing their effect on the protein structure, this study comprehensively analyzed the activation mechanisms of TMEM16F and obtained mechanistic insight into how TMEM16F undergoes a structural change to scramble lipids. However, before publication, several points need to be clarified.

Major points

1. In Fig 1a, fluorescence level was compared between wild-type and mutants in resting state. However, from this data, you cannot judge whether TMEM16F mutants (such as F518A) are activated like a Ca²⁺-stimulated state. Reference fluorescence ratio of stimulation/resting should be indicated.

The values of the initial fluorescence displayed in Fig. 1a strongly correlates with the behavior of the same mutants recorded at elevated Ca²⁺ concentrations. This is illustrated in Supplementary Fig. 1a, where the fluorescence of the same set of cells was measured at elevated intracellular Ca²⁺ concentrations (*i.e.* 760 sec after addition of ionomycin) and is plotted relative to WT. In our study, we intentionally restricted our initial cellular screen of lipid scrambling to the identification of

strongly activating mutants of TMEM16F. For such mutants, we detected appreciable activity already at resting Ca^{2+} concentrations (where WT is essentially inactive). The resulting cumulated exposure of PS on the cell surface is manifested in the increased basal fluorescence level of bound annexin V already prior to the addition of ionomycin (see Figure 1a top). We thus found the initial steady-state fluorescence prior to the elevation of Ca^{2+} as the most sensitive discriminating signal for the detection of activating constructs. Due to the described pronounced fluorescence at resting Ca^{2+} concentrations, the suggested ratio of stimulated/resting fluorescence values is unfortunately not conclusive (as illustrated in Response Fig. 1). We thus prefer to stick to our original representation of activity, since we are convinced that it better illustrates the phenotype of the mutation in our data.

In light of the limitations of this cellular assay of lipid scrambling with respect to the detailed characterization of the Ca^{2+} sensitivity and kinetic properties of mutants, we have turned to an *in vitro* scrambling assay using reconstituted protein to characterize both properties for selected mutants as shown in Fig. 2g, h and Supplementary Fig. 2.

2. In Fig 2e and f, fluorescence level was compared between wild-type and mutants in resting state and activated state. Similar to Fig 1a, the Reference fluorescence ratio of stimulation/resting should be indicated.

See also our comment above. Due to the strongly increased basal fluorescence in mutants with strongly activating phenotype (which includes most investigated mutants of Phe 518), the ratio of initial to activated fluorescence ratio is a poor identifier for their functional behavior. This is evident in the comparison of the fluorescence at resting and elevated Ca^{2+} concentration (as shown in Fig. 2e and f and Response Fig. 1a and b) with the ratio of both values (Response Fig. 1c). Whereas the correlated increase in activity with increasing hydrophilic character of the mutation is evident at low and high Ca^{2+} concentration, the fluorescence ratio decreases with increasing hydrophilicity as consequence of the high basal scrambling activity of the mutants.

Response Fig. 1. Lipid scrambling activity of TMEM16F mutants characterized by a cellular assay. **a, b,** Lipid transport properties of Phe 518 mutants of TMEM16F assayed by the surface staining of cells with fluorescently labeled annexin V. **a,** Initial values recorded at resting Ca²⁺ concentration and **b,** levels measured 600 s after application of ionomycin, which increases intracellular Ca²⁺. **a, b** Data are normalized to the mean value of WT. **c,** Ratio of the fluorescence measured at elevated Ca²⁺ and the initial fluorescence. Bars show mean of indicated individual experiments (depicted as spheres), errors are s.e.m.

3. In Fig 2g, when the authors compare WT to F518H in the reconstituted system, how can the authors determine the reconstitution proportion between these two? If the authors argue that F518H is activated more than WT in 1 mM Ca²⁺-stimulated state, the authors also conclude the and at differences can be observed even in 0 mM Ca²⁺ state?

In our revised manuscript, we show the reconstitution efficiency of WT and mutations of TMEM16F in Supplementary Fig. 2a, b and i. Since in our experience, the pre-prepared liposomes are the most sensitive factor for the reconstitution efficiency of membrane proteins, we have always reconstituted WT and mutants into the same batch of detergent-destabilized liposomes and used only these proteoliposomes for comparative studies. Supplementary Fig 2a, b shows a similar reconstitution efficiency of WT and F518H tested either by Western blot (a) or by size exclusion chromatography after the re-solubilization of reconstituted protein in mild detergents (b). The altered kinetics and increased Ca²⁺ potency of F518H assayed with these characterized proteoliposomes can be appreciated in the raw traces of the scrambling experiments shown in Supplementary Fig. 2c. The reconstitution efficiency of mutants displayed in Supplementary Fig. e-h quantified by SEC is shown in panel i of the same figure.

In contrast to the Ca^{2+} -dependence of scrambling, the rate of basal scrambling at 0 Ca^{2+} is difficult to assay *in vitro*, due to the intrinsic variability of the obtained plateau value even in absence of scrambling (ranging between 40-50% of the initial fluorescence). Whereas Supplementary Fig. 2c shows little difference between WT and F518H at 0 Ca^{2+} , this value varies in another reconstitution of the same mutant shown in Supplementary Fig. 2d. Nevertheless, both reconstitutions show a very similar Ca^{2+} response, where 1 μM Ca^{2+} has a small effect on WT while it nearly fully activates F518H. The difference to the pronounced basal values in cellular assays could be a consequence of the resting Ca^{2+} concentration in cells (which is never zero), the extended amount of time to reach steady state (since constructs show constant basal activity during growth) and differences in the lipid composition of the cellular membrane.

4. In Fig 3a-c, when the authors compare F518H ($\text{Ca}^{2+}(-)$) to WT ($\text{Ca}^{2+}(-)$), how about the structural change?

Our revised manuscript now contains a superposition of F18H^{noCa} and WT^{noCa} (Supplementary Fig. 6a). As expected from the low RMSD (0.64 Å), both structures are very similar. All other comparisons are provided in Fig. 3c, f, i and Supplementary Fig. 6b)

5. In Fig 4a, how the authors can conclude that Nanodisc-reconstituted F518H is an intermediate state? The lipid environment in Nanodisc can be considered a more physiological environment than detergent.

We call the nanodisc reconstituted structure F518H^{Ca}_{ND} a structural intermediate since its coordinates are about halfway on a trajectory between F518H^{Ca} and F518H^{noCa} (as illustrated in Fig. 4a-d and Video 1). Though we do not fully understand the underlying cause for these conformational properties, it is conceivable that the full conformational spread of the protein in nanodiscs is restricted by their limited size and the destabilization of the bilayer at the site of the subunit cavity.

We have clarified this in our manuscript:

Line 185-187:

In this transition, the F518H^{Ca}_{ND} structure shows an apparent intermediate towards the structure of the mutant in detergent since its coordinates are about half way on a potential trajectory from F518H^{noCa} to F518H^{Ca} (Fig. 4a-c).

Line 343-346

While sharing general features, these changes are more extended in the detergent structure F518H^{Ca} than in the nanodisc structure F518H^{Ca}_{ND}, which appears to adopt an intermediate on the transition from F518H^{noCa} to F518H^{Ca} presumably defining the activation process (Fig. 4a-c). Although the large 2N2-MSP nanodiscs provide a condition that is closer to a membrane environment, the accessible conformational space of the protein might be restricted by the size of the disc and the apparent destabilization of the bilayer at the site of the subunit cavity (Fig. 7b), leading to the observed conformational preference.

6. In Fig 5d, WB should be shown with a size marker with molecular weight.

We have added the molecular weight (now Fig. 8d).

Minor points

Consistency on the description in Figures. Ca or Ca²⁺, F518H^{Ca} or N562Ca

We use Ca²⁺ throughout unless it is part of the name of a dataset where Ca is superscripted (F518H^{Ca}).

Reviewer #2 (Remarks to the Author):

The TMEM16 proteins function either as ion channels or lipid scramblase. Among its family members, THEM16F is unique as it can play both roles in conducting ions across the membrane and moving lipid molecules between the two membrane leaflets. Despite extensive studies, including several structures, how the protein performs both roles remains unclear. In the current work, Arndt and colleagues, combining elegant functional assays and structure determination by cryo-electron microscopy, have provided an answer. Using a cellular lipid transport assay and electrophysiology measurements, the authors found that the same mutations

on Helices 4 and 5 affect the activation of both lipid scrambling and ion conductance. Using cryo-EM, the investigators further determined the structures of two of such mutants in the absence and presence of Ca⁺⁺. Comparison of the structures with and without Ca⁺⁺ revealed conformational changes in Helices 4 and 5 that open the ion conductance pore and allow lipid scrambling. Thus, the work has solved a long-standing mystery in the TMEM16 field. The data quality is excellent, and the manuscript is clearly written. The work should certainly be seen by the broad readership of Nature Communications.

One issue that is not discussed, which may be beyond the scope of the current manuscript, is the relationship between ion transport and lipid transport. As both are activated through the same Ca⁺⁺-binding events, is it possible that the two transport processes are coupled? Even a loose coupling mechanism may help lower the energy barrier for transporting charged lipids across the membrane.

Although we do not have data showing a potential coupling between permeating ions and lipids, we do not want to exclude such mechanism. Based on the described data, we think that both conduction processes are mediated by a single protein conformation but proceed at distinct locations in the center of the membrane, where the subunit cavity is occluded from the bilayer. However, since the partial opening of the cavity causing the lipid distortion occurs in close proximity to the ion conduction pore, a mutual influence of both processes leading to some degree of coupling appears plausible. We have briefly mentioned this idea in the discussion. To which extent such mechanism would lower the diffusion of charged lipids remains to be investigated

Line 416-418:

Due to the proximity of their permeation paths, a mutual interaction between both processes appears possible, which could potentially lower the barrier for the diffusion of charged lipids.

1. Page 6, Line 98. To help a reader unfamiliar with the TMEM16 family, the authors can include an overall structure of TMEM16F in Fig. 1 with the parts of interest highlighted.

We have added a panel showing the TMEM16F dimer as Fig. 1c.

2. Page 9, Line 176. F516H(noCa) should be F518H(noCa).

We have corrected the mistake.

3. A scale bar should be included in the electron micrographs, ED Figs. 3a, 4a, 5a, 5b and 7a.

We have included scalebars in the mentioned figure panels.

4. In ED Fig. 3c, 18850 movies should be 18,850 movies.

We have corrected the number.

5. Coloring scheme used in local resolution maps in ED Figs. 3f, 4f, 5g, 6f and 6i is a bit confusing. The color scale should be reversed, with the high resolution (3 Å) areas colored blue and the low resolution (5 Å) red, making it similar to the convention for coloring the B-factor.

We have reversed the color scheme in all indicate panels.

6. In ED Table 1, the number of movies in each dataset should be included.

We have included the number of movies in each dataset.

Reviewer #3 (Remarks to the Author):

This is a remarkable paper offering new insights that illuminate the curious functional differences between TMEM16F, the first scramblase of the TMEM16 family to be identified, and its fungal scramblase counterparts as well as strict ion channel members of the family. The results are based on mutations, tested in cell-free assays, complemented by several cryo-EM structures in detergent and nanodiscs of select mutants. I would avoid the use of the word

catalysis as used in the abstract (line 19) and elsewhere as this has a particular connotation that is not appropriate here.

We have replaced the work catalysis throughout.

I have no major comments except for a clarification (see below).

Figure 1 and associated extended figure 1a are a little confusing. Annexin binding at rest shows that certain mutants have elevated PS exposure – it might be expected that this effect between the different mutants should be damped out on ionomycin treatment where Ca is presumably in excess. Yet F518 stands out under both conditions.

The high value of F518A in our initial screen is partly the consequence of a single high datapoint (these are now explicitly shown in the revised figure). Nevertheless, we find consistently higher fluorescence values under elevated Ca^{2+} concentrations (with a maximum of twice the WT level) also in the more through screen for strongly activating mutations of F518 (*i.e.* F518Q and F518H) shown in Fig. 2f. This indicates that the PS distribution in WT has not yet equilibrated under the applied assay conditions (see also Supplementary Fig. 8a).

REVIEWERS' COMMENTS

Reviewer #1 (Remarks to the Author):

The authors successfully clarified all concerns either through new data, solid explanations, or further discussion. I would strongly recommend the publication of this elegant work to Nature Communications. Congratulation.

Reviewer #2 (Remarks to the Author):

The authors have adequately addressed my concerns in the revised manuscript, and the paper is now ready for publication.

Reviewer #3 (Remarks to the Author):

The revisions have addressed points raised in the initial review.